# Epithelial geometry regulates spindle orientation and progenitor fate during formation of the mammalian epidermis

Kimberly Box, Bradley W Joyce, Danelle Devenport*

Department of Molecular Biology, Princeton University, Princeton, United States

**Abstract** The control of cell fate through oriented cell division is imperative for proper organ development. Basal epidermal progenitor cells divide parallel or perpendicular to the basement membrane to self-renew or produce differentiated stratified layers, but the mechanisms regulating the choice between division orientations are unknown. Using time-lapse imaging to follow divisions and fates of basal progenitors, we find that mouse embryos defective for the planar cell polarity (PCP) gene, *Vangl2*, exhibit increased perpendicular divisions and hyperthickened epidermis. Surprisingly, this is not due to defective Vangl2 function in the epidermis, but to changes in cell geometry and packing that arise from the open neural tube characteristic of PCP mutants. Through regional variations in epidermal deformation and physical manipulations, we show that local tissue architecture, rather than cortical PCP cues, regulates the decision between symmetric and stratifying divisions, allowing flexibility for basal cells to adapt to the needs of the developing tissue.

DOI: https://doi.org/10.7554/eLife.47102.001

## Introduction

Oriented cell division is a fundamental mechanism through which multicellular organisms build complex tissue architecture (*Bergstrahl et al., 2017*; *Gillies and Cabernard, 2011*). By controlling the angle of the mitotic spindle, a cell can position its daughters to facilitate tissue expansion, establish multiple cell layers, or generate asymmetric cell fates. Two primary mechanisms direct the orientation of cell division: intrinsic cues that couple the mitotic spindle to cell polarity, and extrinsic factors that guide spindle alignment relative to physical or chemical cues (*di Pietro et al., 2016*). Although proper tissue assembly requires hard-wired spindle orienting programs to reproducibly build functional organs, cells also need flexibility to divide according to changing needs of the tissue. The mechanisms that ensure flexibility in spindle orientation during development are poorly understood.

The murine epidermis is a well-studied model to investigate the regulation of spindle orientation during development. The formation of the skin's stratified layers, which serve as a barrier to protect the organism from its external environment, relies on oriented cell divisions. Unlike simple epithelia, which divide exclusively within the epithelial plane, cells of the innermost, basal layer of the epidermis divide in both planar and perpendicular orientations (*Lechler and Fuchs, 2005*; *Poulson and Lechler, 2010*; *Smart, 1970*). Perpendicularly oriented cell divisions generate one basal and one suprabasal daughter cell that goes on to differentiate and contribute to stratification of the tissue. Cell divisions that occur parallel to the epithelial plane generate two basal daughters, which facilitates tissue growth and expansion of the basal progenitor pool (*Ray and Lechler, 2011*). The ability of basal epidermal cells to conduct both types of division orientations is imperative for the proper formation and maintenance of the tissue. Too many perpendicular divisions can deplete the progenitor pool while too many symmetric divisions leads to defective barrier formation (*Niessen et al.,*

*For correspondence:
danelle@princeton.edu

**Competing interests:** The authors declare that no competing interests exist.

*2013*; *Williams et al., 2011*). How the balance between symmetric and differentiating divisions is regulated during formation of the epidermis is not understood.

During embryonic development, the epidermis transforms from a single layer of basal progenitors at E12.5 to a stratified and differentiated epithelium with barrier function by E17.5 (*Hardman et al., 1998*; *Kulukian and Fuchs, 2013*; *Muroyama and Lechler, 2012*). Stratification occurs in at least two stages, an early phase between E13.5-E15.5 where oblique and perpendicular divisions produce the first few stratified layers, and a late phase at E16.5-E17.5 where perpendicular divisions and differentiation drive epidermal maturation (*Lechler and Fuchs, 2005*; *Smart, 1970*; *Williams et al., 2011*; *Williams et al., 2014*). Intrinsic polarity cues orient these late stage perpendicular divisions through the spindle anchoring proteins LGN and Inscuetable (mInsc). Apical polarity factors including Par3 recruit the apical localization of mInsc, which induces the localization of Gαl, LGN, and NuMA, a conserved protein complex that anchors spindle astral microtubules to the cell cortex (*Lechler and Fuchs, 2005*; *Poulson and Lechler, 2010*; *Williams et al., 2011*; *Williams et al., 2014*). Surprisingly, these factors are not required during the early phase of stratification (*Williams et al., 2014*). What regulates division orientations early in epidermal development remains unknown. Moreover, the cues that direct planar division orientations in the epidermis have not been identified.

Because apical-basal polarity cues direct perpendicular divisions to drive stratification, we hypothesized that parallel, symmetric cell divisions might be oriented by planar cell polarity (PCP), which controls polarization along an epithelial plane (*Butler and Wallingford, 2017*; *Devenport, 2016*; *Goodrich and Strutt, 2011*; *Segalen and Bellaïche, 2009*). In the epidermis, a conserved set of PCP components including the transmembrane proteins Celsr1, Frizzled-6 (Fz6) and Van Gogh-like 2 (Vangl2), localize to basolateral cell junctions preferentially along anterior-posterior edges (*Aw et al., 2016*; *Devenport and Fuchs, 2008*; *Devenport et al., 2011*). PCP protein asymmetry coincides with the onset of stratification, and we had previously noted that embryos harboring a homozygous point mutation in the *Vangl2* gene (called *Looptail (Lp); Vangl2$^{Lp/Lp}$*) display increased epidermal thickness, suggesting a role for PCP in planar division orientations (*Devenport and Fuchs, 2008*). However, the cellular basis of this epidermal thickening and its emergence over time had not previously been investigated.

Using live imaging to trace the orientation and positional fate of basal cell divisions, we investigate the function of PCP and tissue mechanics early in epidermal stratification. In agreement with the hypothesis that PCP promotes planar basal cell divisions, we find that Vangl2 mutant embryos display an increased proportion of stratifying divisions at the expense of planar divisions. Unexpectedly, this is not a direct result of defective PCP; rather, we show that failure of the epidermis to close over the neural tube in these mutants leads to cell crowding within the basal layer, concomitant with changes in cell width and height and a higher frequency of asymmetric cell divisions. Combining our live imaging approach with tissue-specific genetic perturbations and mechanical tissue manipulations, we show that early in epidermal development, interphase 3D cell geometry and packing, rather than cortical PCP cues, influence mitotic spindle orientation. Although it is well established that cell divisions align with the interphase long axis in two dimensions, our data show that this mechanism can also impact spindle alignment in the third, apical-basal dimension in a stratifying epithelium. We propose that by responding to changes in cell density and shape, basal cell division orientations can adjust to satisfy the needs of the tissue.

## Results

### Basal progenitor cells in *Vangl2$^{Lp/Lp}$* embryos undergo fewer planar cell divisions

The relationship between division orientation and epidermal stratification has been primarily inferred from observations made in fixed tissue sections. This method comes with some key disadvantages: it excludes divisions oriented orthogonal to the plane of sectioning and does not allow one to directly correlate mitotic spindle alignment with the final positions of daughter cells. To overcome these limitations, we performed time-lapse imaging of live, flat mounted skin explants expressing a nuclear marker, K14-H2B-GFP (*Tumbar et al., 2004*), which allowed us to observe cell divisions dynamically in three-dimensional space (*Figure 1*, *Figure 1—video 1* and *Figure 1—video 2*). Z-stacks were

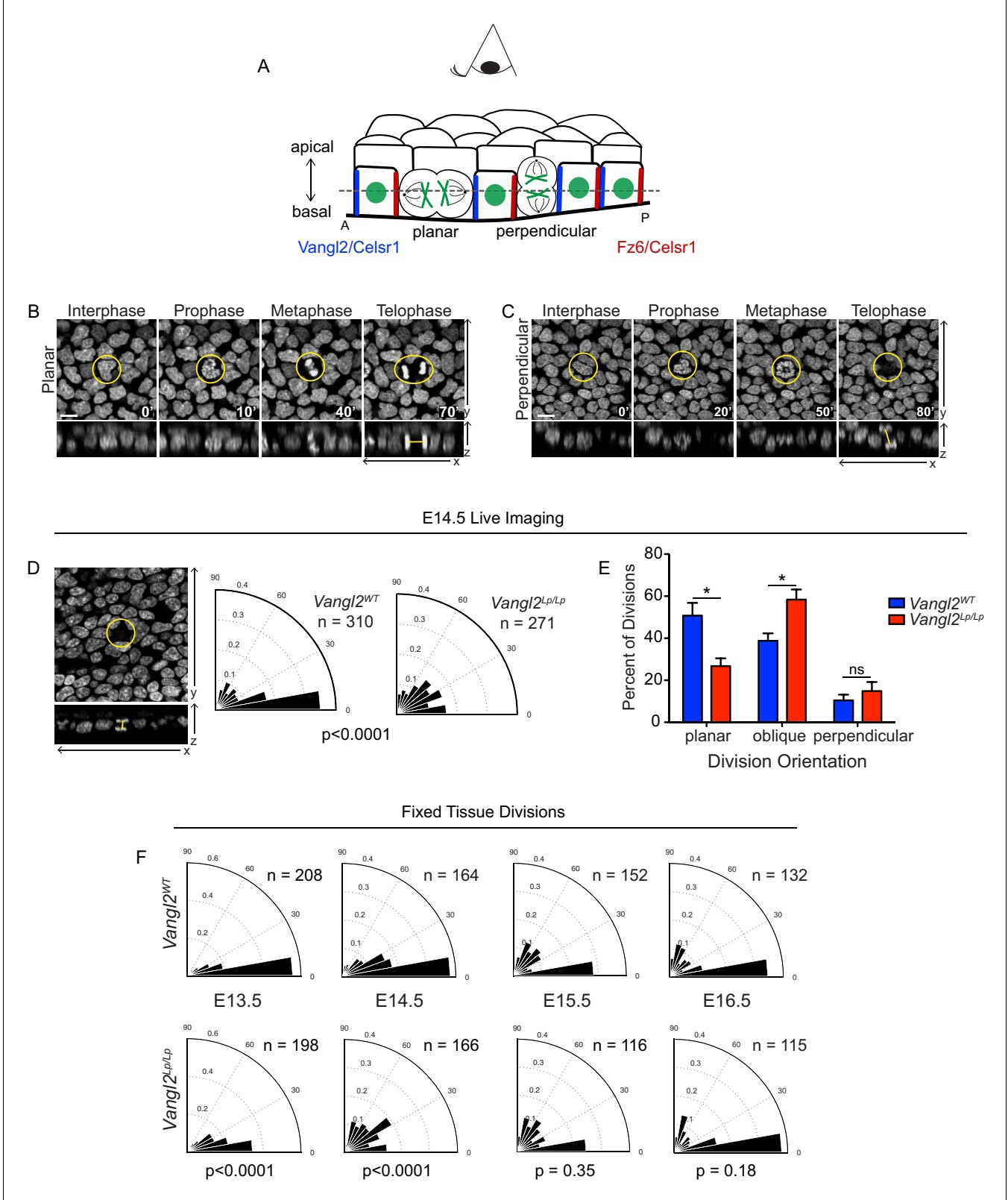

**Figure 1.** Basal cell division orientations in wildtype and Vangl2 mutant embryos. (**A**) Schematic of E14.5 skin depicting several planar cell polarity components and example division orientations. Dotted line represents focal plane for live imaging. (**B,C**) Stills from time-lapse movies of *Vangl2^WT* E14.5 skin explants expressing K14-H2B-GFP, showing examples of planar (**B**) and perpendicular (**C**) division orientations. Top panels are the planar view of the basal layer of the epidermis, bottom panels are XZ dimension. Scale bar = 10 μm. See also *Figure 1—video 1* and *Figure 1—video 2*. (**D**)
*Figure 1 continued on next page*

*Figure 1 continued*

Example and quantification of division angles in live epidermal explants at E14.5. *Vangl2^{WT}*, n = 310 divisions pooled from three embryos. *Vangl2^{Lp/Lp}*, n = 271 divisions from three embryos. Modified Kuiper's Test, p=1.7523e-15. (E) Distribution of division orientations in live epidermal explants at E14.5. Planar: Θ <= 20°, unpaired two-tailed t-test p=0.027; oblique: 20°>Θ>=70°, p=0.030; perpendicular: 70°>Θ>=90°, p=0.427. n = 3 explants from each genotype. (F) Angular frequency of division angles quantified from fixed whole mount skins dissected from embryos e13.5 – e16.5. E13.5: *Vangl2^{WT}*, n = 208 divisions from three embryos; *Vangl2^{Lp/Lp}*, n = 198 divisions from two embryos; p=2.8648e-08. E14.5: *Vangl2^{WT}*, n = 164 divisions from three embryos; *Vangl2^{Lp/Lp}*, n = 166 divisions from three embryos; p=1.5118e-18. E15.5: *Vangl2^{WT}*, n = 152 divisions from three embryos; *Vangl2^{Lp/Lp}*, n = 116 divisions from three embryos; p=0.3482. E16.5: *Vangl2^{WT}*, n = 132 divisions from three embryos; *Vangl2^{Lp/Lp}*, n = 115 divisions from three embryos; p=0.1785.

DOI: https://doi.org/10.7554/eLife.47102.002

The following videos are available for figure 1:

**Figure 1—video 1.** Planar view of a basal cell from an E14.5 *Vangl2^{WT}* skin explant expressing K14-H2B-GFP undergoing a planar division.

DOI: https://doi.org/10.7554/eLife.47102.003

**Figure 1—video 2.** Example from an E14.5 *Vangl2^{Lp/Lp}* skin explant expressing K14-H2B-GFP showing a basal cell undergoing a perpendicular division.

DOI: https://doi.org/10.7554/eLife.47102.004

captured every 10 min over approximately 8 hours of epidermal development, and division planes were quantified by measuring the angle between the centroids of the two daughter nuclei during telophase. Divisions were categorized as planar (Θ<=20°), oblique (20<Θ<=70°) or perpendicular (70°>Θ>=90°) depending on their orientation relative to the plane of the basal layer. Interestingly, chromatids that were aligned along the metaphase plate continued to rotate prior to separating at anaphase/telophase. Because of this instability in spindle alignment during metaphase, division planes were measured only after chromatid separation.

In wildtype explants at E14.5, approximately half of all basal cell divisions were oriented parallel to the epithelial plane (51 ± 6%). However, this proportion was reduced in explants from *Vangl2^{Lp/Lp}* mutants (27 ± 4% planar divisions), which exhibited an increased frequency of oblique and perpendicular divisions (*Figure 1D,E*). Importantly, division orientations observed by live imaging mirrored the distribution in fixed whole mount epidermis, confirming that our explant imaging system reflects division behavior in vivo (*Figure 1F*). The altered distribution of division orientations at E14.5 suggested a potential role for PCP in promoting planar basal cell divisions in the epidermis.

To explore trends in cell division orientation over developmental time, we quantified division planes in fixed whole mount skins across multiple time points from E13.5-E16.5 (*Figure 1F*). Consistent with previous reports, we observed a strong bias toward planar division orientations in wildtype embryos at E13.5 when the epithelium is predominantly comprised of a single layer (*Lechler and Fuchs, 2005*; *Williams et al., 2014*). At E15.5 and E16.5, the proportion of oblique and perpendicular divisions increased, coinciding with the expansion of stratified epidermal layers. However, our data show a much higher proportion of planar and oblique divisions at E16.5 compared to prior reports, which we attribute to differences in observing divisions in whole mount versus sagittal tissue sections, as planar and oblique divisions that align out of the plane of sectioning are likely to go uncounted (*Asare et al., 2017*; *Lechler and Fuchs, 2005*; *Luxenburg et al., 2011*; *Niessen et al., 2013*; *Poulson and Lechler, 2010*; *Williams et al., 2011*; *Williams et al., 2014*). Interestingly, Vangl2 mutant embryos displayed elevated oblique and perpendicular divisions at E13.5 and E14.5 but not at later embryonic stages, suggesting the requirement for PCP function in promoting planar divisions is restricted to an early developmental window of time corresponding to the onset of epidermal stratification.

## Division plane is a strong predictor of the positional fate of daughter cells

After each cell division in the epidermis, the final fates and positions of the two daughter cells influence the stratified architecture of the developing skin. Daughter cells with suprabasal positioning differentiate and contribute to the skin's upper stratified layers. Daughters with basal positioning retain progenitor fate and continue to divide. Previously, the fates of daughter cells were inferred by their division planes as measured in fixed tissue sections. To directly assess how a basal cell's division plane relates to its positional fate within the tissue, we employed our live imaging system to track recently divided cells and assign each cell a final positional fate. Fates were defined as symmetric

when both daughter cells remained in the basal layer for at least 1.5 hr following division. Asymmetric fates were scored when one daughter cell remained basal while the other cell moved suprabasally (*Figure 2A–C*). We found that in wildtype embryos at E14.5 division orientation and cell fate were tightly linked. For example, 94% of cells that divided within 20 degrees of the epithelial plane adopted symmetric fates (*Figure 2A*, blue points). Similarly, 89% of cells that divided within 70–90 degrees of the plane adopted asymmetric fates (*Figure 2A*, green points). Daughter cells resulting from oblique divisions, by contrast, had a roughly equal chance of becoming symmetric or asymmetric (41% symmetric versus 59% asymmetric) (*Figure 2A*, magenta points). In *Vangl2$^{Lp/Lp}$* embryos, however, the relationship between division orientation and final position was shifted toward asymmetric. Only 78% of planar divisions and 21% of oblique divisions resulted in symmetric positional fates (*Figure 2B,C*).

Notably, we did not observe delamination of basal cells into the suprabasal layer. Although we were not specifically seeking to quantify delamination events in this study, it is striking that we did not observe them in our time-lapse experiments. In contrast to previous studies suggesting early epidermal stratification is driven predominantly by basal cell delamination (*Miroshnikova et al., 2018*; *Wickström and Niessen, 2018*; *Williams et al., 2014*), our results suggest that early stratification results from oblique and perpendicular divisions that generate suprabasal daughters.

To investigate how altered division planes contribute to skin structure in PCP mutants, we quantified epithelial thickness over the course of epidermal development. The epithelium of Vangl2 mutants was measurably thicker than control embryos as early as E15.5, and this difference increased by E16.5. At E18.5, Vangl2 mutant embryos continue to display significantly thicker skin than their wildtype littermates, suggesting the early bias toward oblique and perpendicular divisions, together with a preference for asymmetric cell fates, may contribute to longer-term effects on epidermal structure (*Figure 2E–H*).

## Early in epidermal development, LGN localization does not correlate with planar spindle positioning

Previous work exploring the mechanisms underlying oriented cell division in the murine epidermis has shown that late in epidermal development, apical-basal polarity factors localize several spindle-orienting components, such as LGN. These components serve to position the mitotic spindle to promote perpendicular, asymmetric basal cell divisions (*Lechler and Fuchs, 2005*; *Williams et al., 2011*; *Williams et al., 2014*). We reasoned that PCP factors might recruit these components to the lateral cortex to orient planar divisions, and that a failure of LGN to localize laterally might explain the increased oblique and perpendicular divisions observed in Vangl2 mutants. After confirming prior reports of apical LGN localization in cells with perpendicularly-oriented spindles at E16.5 (*Figure 3—figure supplement 1*), we characterized the localization of this protein in dividing cells at E14.5. At this earlier stage, we observed apical LGN in a subset of cells undergoing oblique and perpendicular divisions. However, in cells undergoing planar divisions, LGN exhibited lateral localization in only 14% of cells, and was most frequently undetectable (61% of planar divisions; *Figure 3A,B*). These patterns of LGN localization are consistent with those previously observed in thin tissue sections by Williams et al. together with the observation that LGN knockdown leads to an increase in planar divisions (*Williams et al., 2014*), these data suggest that this spindle-orienting pathway is not the primary mechanism promoting planar basal cell divisions.

We also explored the possibility that the observed increase in oblique/perpendicular divisions and asymmetric cell fates in *Vangl2$^{Lp/Lp}$* mutants is caused by more frequent apical LGN localization. Although the occurrence of apical LGN in *Vangl2$^{Lp/Lp}$* embryos (47.7% of divisions) was slightly increased compared to *Vangl2$^{WT}$* (41.1% of divisions), this increase is not sufficient to explain the altered division orientations (*Figure 3C,D*), suggesting that Vangl2 does not act through LGN to orient divisions. Together, these data are consistent with a potential role for Vangl2 in an alternative mechanism that influences division planes during the early phase of epidermal stratification.

## Cell division orientation is correlated with basal cell geometry

Since division planes at E14.5 did not correlate with localization of a known epidermal spindle-orienting cue, we hypothesized that basal cells might instead divide according to cell geometry. Hertwig's rule, also known as the 'long axis rule', states that a cell is most likely to divide along its longest

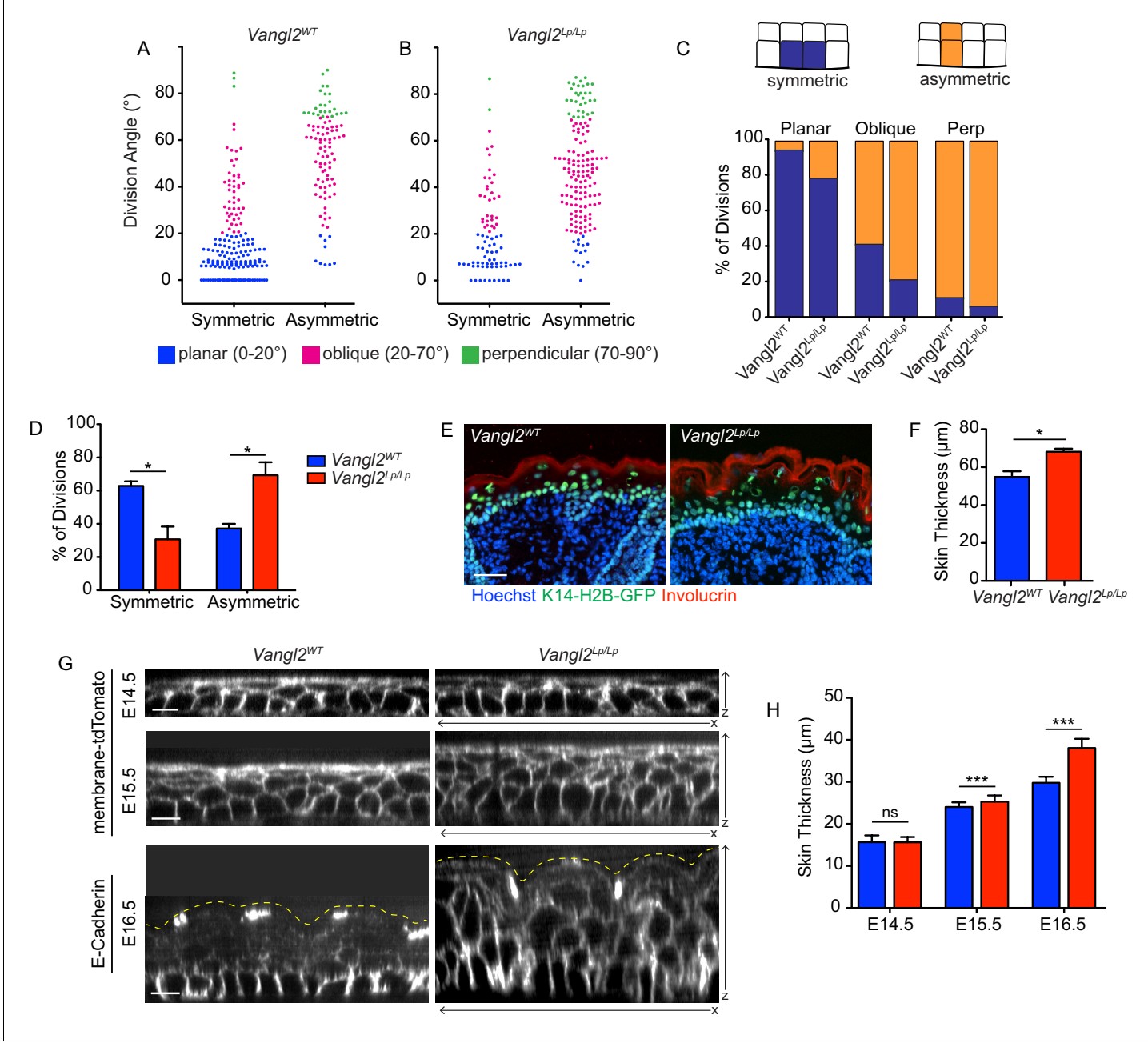

**Figure 2.** Positional cell fates correlate with division orientation. (**A, B**) The relationship between division angle and final cell positions. Daughter cells were followed for 1.5 hr after cell division, and assigned a positional fate: symmetric or asymmetric. Each dot represents a single division event. (**C**) Representation of the same data in (**A,B**), showing the distribution of fates for each division orientation. For planar divisions: symmetric fates = 94% [*Vangl2^WT*], 78% [*Vangl2^Lp/Lp*]. Oblique divisions: symmetric fates = 41% [*Vangl2^WT*], 21% [*Vangl2^Lp/Lp*]. Perpendicular divisions: asymmetric fates = 89% [*Vangl2^WT*], 94% [*Vangl2^Lp/Lp*]. (**D**) Combining all division orientations, *Vangl2^Lp/Lp* embryos display an overall bias toward asymmetric final cell positions. Unpaired two-tailed t-test, p=0.017. For (**A–D**), n = 284 divisions pooled across three *Vangl2^WT* embryos; n = 238 divisions pooled from three *Vangl2^Lp/Lp* embryos. (**E**) Sagittal sections of E18.5 skin from *Vangl2^WT; K14-H2B-GFP* and *Vangl2^Lp/Lp; K14-H2B-GFP* embryos. Involucrin (red) labels the outer stratified layers and nuclei are labeled with Hoechst. Scale bar = 50 μm. (**F**) Quantification of skin thickness at E18.5. n = 10 images for each of three embryos per genotype. Bars represent means of the three embryos, error bars are SEM. Unpaired two-tailed t-test, p=0.017. (**G**) XZ panels from whole mount images of *Vangl2^WT* and Vangl2 mutant skins at E14.5 – E16.5, expressing membrane-tdTomato or immunostained for E-Cadherin. Yellow dotted lines outline the outermost epidermal layer. Scale bars = 10 μm. (**H**) Quantification of skin thickness at E14.5 – E16.5. n = 30 measurements per genotype per stage. Bars represent means, error bars are SD. E14.5: unpaired two-tailed t-test, p=0.9340; E15.5: unpaired two-tailed t-test, p=0.0003; E16.5: unpaired two-tailed t-test, p<0.0001.

DOI: https://doi.org/10.7554/eLife.47102.005

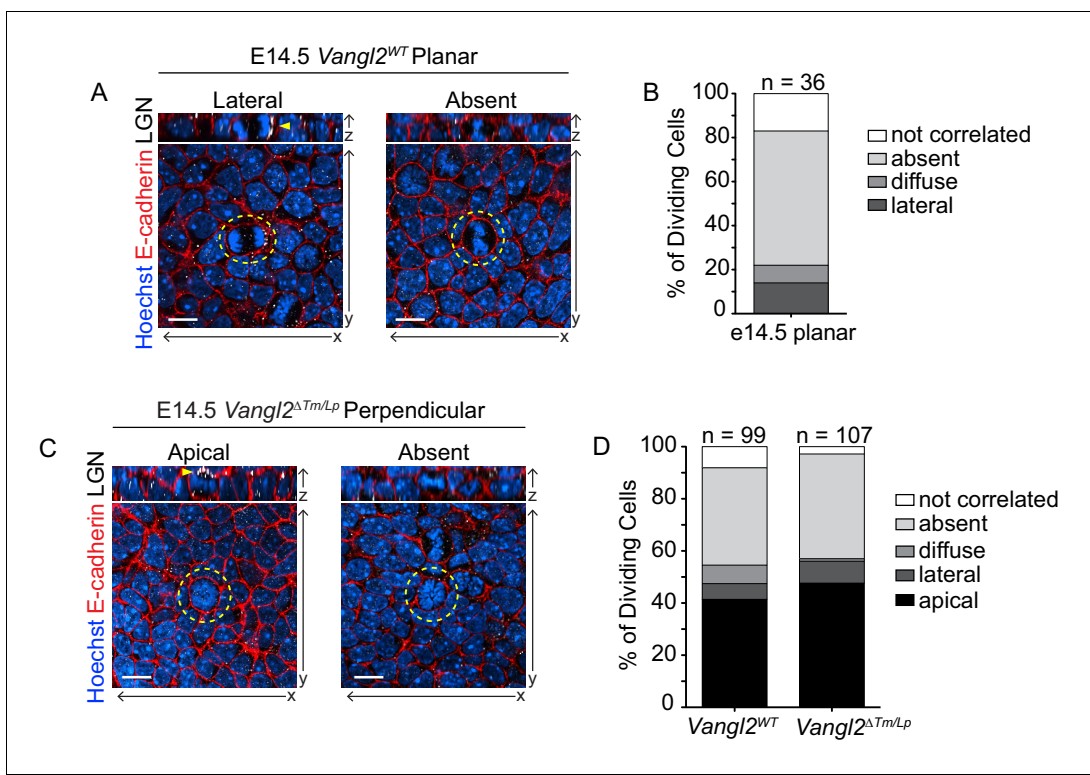

**Figure 3.** LGN localization does not determine planar division orientation. (**A**) Examples from *Vangl2^WT^* embryos of cells dividing in a planar orientation with unilateral (left) and absent (right) LGN localization. In all images, Hoechst labels the nuclei, cell membranes are marked with E-Cadherin (red) and LGN is white. Yellow dashed lines outline the dividing cell, and yellow arrowheads mark LGN localization. (**B**) Frequency of LGN localization patterns in planar cell divisions in *Vangl2^WT^* embryos at E14.5. n = 36 cells (metaphase-telophase). 'Not correlated' LGN refers to a detectable signal that does not correlate with division plane (i.e., basal in a perpendicularly dividing cell). (**C**) Examples from *Vangl2^ΔTm/Lp^* embryos of cells dividing in a perpendicular orientation with apical (left) and absent (right) LGN. (**D**) Frequency of LGN localization patterns in all dividing cells in *Vangl2^WT^* and *Vangl2^ΔTm/Lp^* embryos at E14.5. n = 99 and 107 cells, respectively (metaphase-telophase). All scale bars = 10 μm.
DOI: https://doi.org/10.7554/eLife.47102.006

The following figure supplement is available for figure 3:

**Figure supplement 1.** Apical LGN localization at E16.5.
DOI: https://doi.org/10.7554/eLife.47102.007

interphase axis, and has been observed in many cell types dividing within a two-dimensional plane (*Bosveld et al., 2016*; *Théry and Bornens, 2006*; *Théry et al., 2005*; *Wyatt et al., 2015*). In the epidermis, where basal cells divide in three-dimensional space, a cell's longest axis may lie perpendicular to the epithelial plane, which could promote perpendicular spindle alignment and asymmetric division. Indeed, the perpendicular divisions of early *Xenopus* and zebrafish embryos have been shown to divide according to a three-dimensional long-axis rule (*Chalmers et al., 2003*; *Xiong et al., 2014*). Interestingly, we noted that basal cells appeared taller and narrower in Vangl2 mutant embryos (*Figure 4A*), which, according to Hertwig's rule, might account for the observed increase in asymmetric divisions. To test this idea we first quantified basal cell area and height in fixed whole mount explants from E14.5 wildtype and *Vangl2^Lp/Lp^* embryos. Compared to *Vangl2^WT^* embryos, basal cells in *Vangl2^Lp/Lp^* mutants were significantly smaller in cross-sectional area along the planar axis and taller along the apical-basal axis (*Figure 4A–C*). This increase in height:width ratio was not accompanied by changes in the number of neighbors, as *Vangl2^Lp/Lp^* and wildtype embryos displayed a similar distribution of cells with 4,5,6,7, and eight edges (*Figure 4D*). Basal cells in *Vangl2^Lp/Lp^* embryos were packed more densely into the basal layer compared to wildtype (average = 258 cells vs 215 cells per 1200 μm² field of view) (*Figure 4E*). However, we did not detect elevated proliferation in *Vangl2^Lp/Lp^* embryos (*Figure 4F*). Thus, the geometrical differences observed

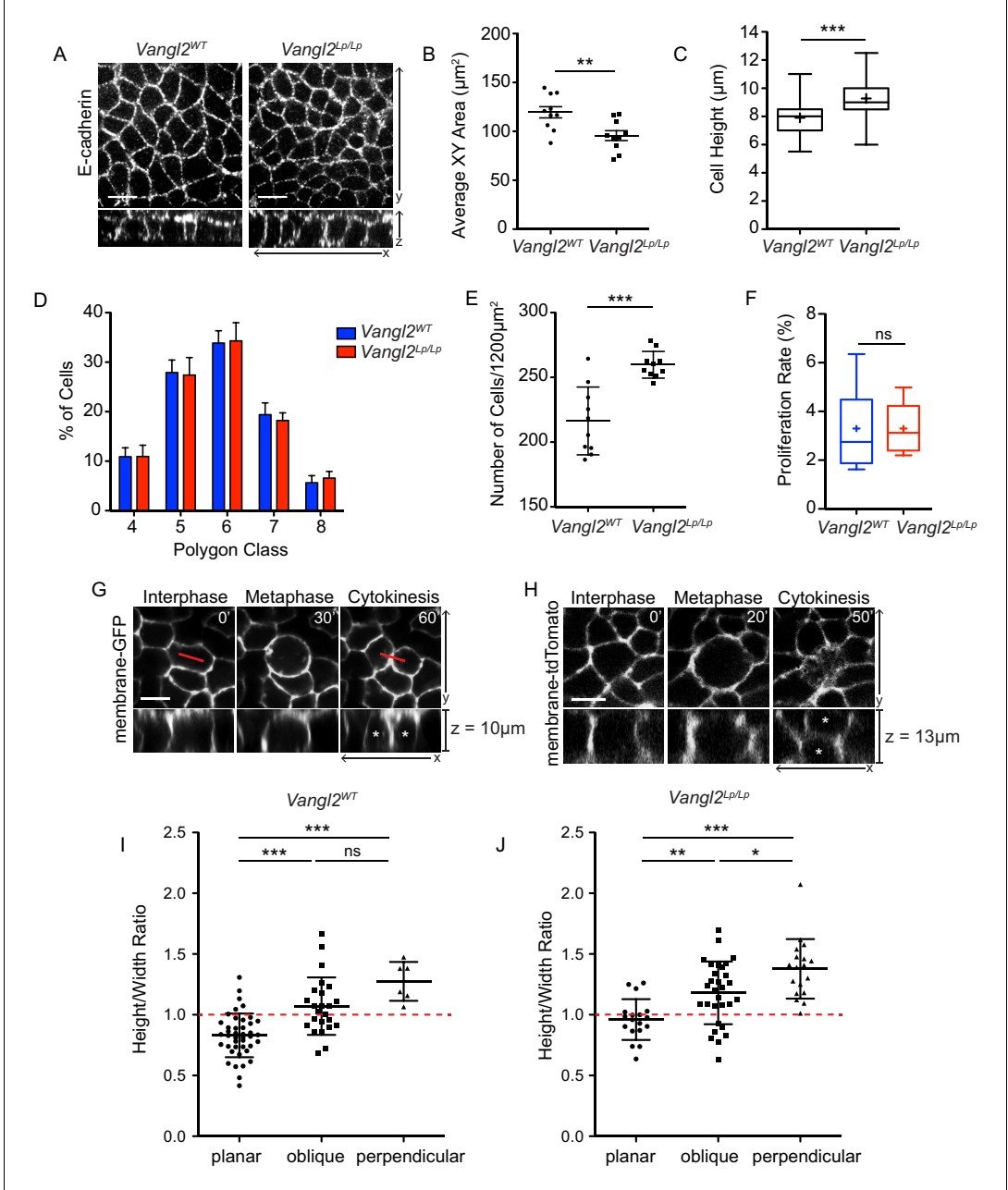

**Figure 4.** Cell division orientation correlates with basal cell geometry. (**A**) Representative images of basal layer labeled with E-Cadherin from E14.5 *Vangl2^WT^* and *Vangl2^Lp/Lp^* embryos. (**B**) Quantification of cross sectional areas of basal cells. Each dot represents average area of all cells in a single field of view. n = 10 total fields of view from three embryos per genotype. Bars are mean with SEM. Unpaired two-tailed t-test, p=0.0057 (**C**) Quantification of cell height along apical-basal axis. 300 cells were measured from across three embryos per genotype. Whiskers indicate minimum and maximum values, + indicates the mean. Unpaired two-tailed t-test, p<0.0001. (**D**) Quantification of distribution of polygon classes, as defined by the number of each cell's neighbors. n = 10 total fields of view from three embryos per genotype. Error bars denote SD. (**E**) Quantification of cell density, as number of cells per field of view (1200 μm²). Each dot represents a single field of view. n = 10 fields of view across three embryos per genotype. Bars are mean with SD. Unpaired two-tailed t-test, p=0.0001. (**F**) Proliferation rates, quantified as the number of mitotic cells in a field of view as a percentage of total number of cells. n = 10 fields of view across three embryos. Whiskers denote min-max values, mean shown as '+.' Unpaired two-tailed t-test, p=0.9987. (**G**) Still images from a time-lapse movie of E14.5 *Vangl2^WT^; K14-Cre; mTmG* explant showing an example of a basal cell dividing within the plane of the epidermis, along its longest interphase axis. Red line denotes orientation of the longest XY axis of the cell. See also ***Figure 4—video 1***. (**H**) Still images from a time-lapse movie of an E14.5 skin explant from *Vangl2^ΔTm/Lp^; mTmG* embryo showing a cell dividing in a perpendicular orientation. Daughter cells are marked with asterisks in the XZ view. See also ***Figure 4—video 2***. (**I,J**) Relationship between height:width aspect ratio and division orientation. H:W ratios were measured from interphase cells at the time just before the onset of mitotic rounding. Width is defined as the

*Figure 4 continued on next page*

*Figure 4 continued*

longest planar axis of the cell. Each dot corresponds to a single division event, and divisions were binned according to the angle of the division plane in cytokinesis. Bars are mean with SD. (I) n = 72 divisions pooled from three E14.5 embryos. One-way ANOVA, p<0.0001. Tukey's Multiple Comparison Test: planar vs oblique, p<0.05; planar vs perpendicular, p<0.05; oblique vs perpendicular, p>0.05. (J) n = 66 divisions from two E14.5 $Vangl2^{\Delta Tm/Lp}$ embryos. One-way ANOVA, p<0.0001. Tukey's Multiple Comparison Test: planar vs oblique, p<0.05; planar vs perpendicular, p<0.05; oblique vs perpendicular, p<0.05. All scale bars = 10 μm.

DOI: https://doi.org/10.7554/eLife.47102.008

The following video and figure supplements are available for figure 4:

**Figure supplement 1.** Characterization of epidermal cell shape and denisty at E13.5 and E15.5.

DOI: https://doi.org/10.7554/eLife.47102.009

**Figure supplement 2.** Correlation between division angle and height:width ratio.

DOI: https://doi.org/10.7554/eLife.47102.010

**Figure 4—video 1.** Planar view of a basal cell from an E14.5 $Vangl2^{WT}$ skin explant expressing membrane-GFP, undergoing a planar cell division along its longest interphase axis.

DOI: https://doi.org/10.7554/eLife.47102.011

**Figure 4—video 2.** Planar view of a basal cell from an E14.5 $Vangl2^{\Delta Tm/Lp}$ skin explant expressing membrane-tdTomato, undergoing a perpendicular cell division.

DOI: https://doi.org/10.7554/eLife.47102.012

in $Vangl2^{Lp/Lp}$ mutants are defined not by differences in cell sidedness, but rather by a change in height:width aspect ratio. We observed similar altered cell geometries in Vangl2 mutants at E13.5, when the proportion of oblique and perpendicular divisions is also increased. However, we did not observe such differences at E15.5, when division planes are similar to wildtype (*Figure 4—figure supplement 1*), perhaps because elevated asymmetric divisions at earlier stages relieves basal cell crowding. Together these data show that early in epidermal development, $Vangl2^{Lp/Lp}$ cells are more crowded and elongated along their apical-basal axes and, according to Hertwig's rule, may be more likely to divide in an oblique or perpendicular orientation.

After observing this tissue-wide correlation between 3D cell geometry and cell division orientations, we wanted to more directly assess this relationship on a per-cell basis. We therefore performed live-imaging of E14.5 skin explants expressing either a membrane-GFP or membrane-tdTomato marker (*Muzumdar et al., 2007*), which allowed us to measure cell geometry in interphase and observe that same cell's division orientation. Confirming what has previously been shown for simple epithelia dividing in a two-dimensional plane, we found that basal cells dividing within the plane of the epidermis do so along their longest XY axis (*Figure 4—video 1*). We also observed a modest but significant relationship between division plane and height:width aspect ratio along the apical-basal axis, wherein cells that divided in a planar orientation tended to have smaller height:width ratios (average = 0.8307) compared to cells that divided perpendicularly (average height:width ratios = 1.275) (*Figure 4I—figure supplement 2*). This relationship was also observed in Vangl2 mutant embryos (*Figure 4H,J—figure supplement 2*, *Figure 4—video 2*), where perpendicular divisions were associated with larger height:width ratios. These results suggest that during the early phase of epidermal stratification, the decision to divide in a planar or perpendicular orientation is influenced by a cell's three-dimensional geometry. Furthermore, these results also show that basal cells can still divide according to their geometrical state in the absence of PCP function.

## Cell shape and division orientation defects in Vangl2 mutants are indirect, and likely a consequence of improper neural tube closure

The altered cell geometries and bias toward oblique and perpendicular cell divisions observed in $Vangl2^{Lp/Lp}$ embryos could be directly or indirectly influenced by PCP function. As planar cell polarity components are highly expressed in the basal layer, they could have a direct role in regulating basal cell shape. Alternatively, changes in epidermal cell shape could arise as a secondary consequence of the open neural tube phenotype characteristic of several PCP mutants (*Curtin et al., 2003*; *Hamblet et al., 2002*; *Kibar et al., 2001*; *Murdoch et al., 2001*; *Wang et al., 2006*). The process of neural tube closure pulls the adjoining surface ectoderm over the dorsal midline to cover the spinal cord. When this process is disrupted, as in Vangl2 knockouts, the epidermis fails to enclose the embryo. To distinguish between direct and indirect functions for PCP in epidermal stratification, we

conditionally deleted Vangl1 and Vangl2 in the epidermis using K14-Cre (*K14-Cre; mTmG; Vangl1*[fl/fl]*; Vangl2*[fl/fl] hereafter referred to as *Vangl1,2 dcKO*) (*Vasioukhin et al., 1999*). These embryos complete proper neural tube closure, but display severe planar cell polarity defects in the epidermis; PCP proteins fail to asymmetrically localize in the basal layer (*Figure 5A*) and hair follicles emerge with vertical instead of anterior-directed growth (*Cetera et al., 2017*; *Chang et al., 2016*). Nevertheless, in *Vangl1,2 dcKO* embryos, the height and cross-sectional area of basal cells were comparable to control littermates (average surface area = 116.5 μm$^2$ vs 116.8 μm$^2$; average height = 7.89 μm vs 7.84 μm for control and *Vangl1,2 dcKO*, respectively), and cell density was unchanged (*Figure 5B–D*). Moreover, the distribution of division orientations in *Vangl1,2 dcKO* embryos was similar to controls (*Figure 5E,F*), suggesting that the phenotypes observed in the *Vangl2*[Lp/Lp] embryos are a result of failed neural tube closure, and not directly due to the absence of Vangl2 or planar cell polarity function in the epidermis.

We also quantified division planes and cell geometries in embryos lacking Frizzled-6, which display strong planar cell polarity defects in the skin (*Cetera et al., 2017*; *Chang et al., 2016*;

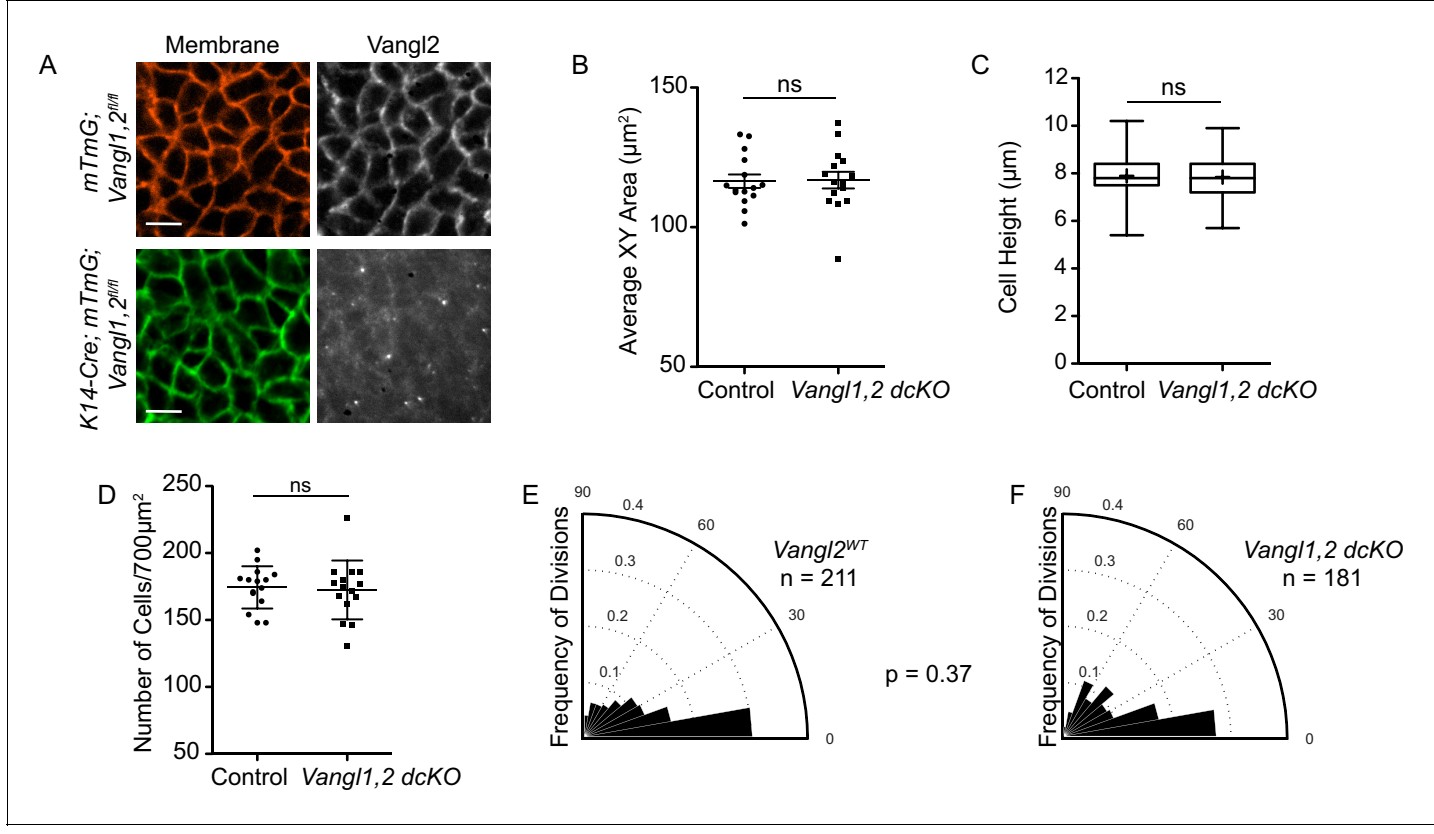

**Figure 5.** PCP mutants that undergo proper neural tube closure do not display altered cell shapes or division orientations. (**A**) Representative images of the basal layer of *Vangl1*[fl/fl]*; Vangl2*[fl/fl]*; mTmG* control and *Vangl1*[fl/fl]*, Vangl2*[fl/fl]*; K14-Cre; mTmG* embryos. Brightness of Vangl2 panel in *Vangl1*[fl/fl]*; Vangl2*[fl/fl]*; K14-Cre; mTmG* was increased to show lack of Vangl2 staining. Scale bars = 10 μm. (**B**) Quantification of basal cell cross-sectional areas. Each dot represents average area of all cells in a single field of view. n = 15 fields of view across three embryos per genotype. Bars are mean with SEM. Unpaired two-tailed t-test, p=0.9209. (**C**) Quantification of cell height along apical-basal axis. n = 300 cells per genotype. Whiskers indicate minimum and maximum values, + indicates the mean. Unpaired two-tailed t-test, p=0.4209. (**D**) Quantification of cell density, as number of cells per field of view (700 μm$^2$). Each dot represents a single field of view. n = 15 fields of view across three embryos per genotype. Bars are mean with SD. Unpaired two-tailed t-test, p=0.7840. (**E**) Division orientations in *Vangl1*[fl/fl]*; Vangl2*[fl/fl]*; mTmG* control embryos. n = 211 divisions, from 45 images across three embryos. (**F**) Division orientations in *Vangl1*[fl/fl]*; Vangl2*[fl/fl]*; K14-Cre; mTmG* embryos. n = 181 divisions, from 45 images across three embryos. Modified Kuiper's Test, p=0.3743.

DOI: https://doi.org/10.7554/eLife.47102.013

The following figure supplement is available for figure 5:

**Figure supplement 1.** Basal cell shape and division orientations are unaffected in Fz6 knockout embryos.

DOI: https://doi.org/10.7554/eLife.47102.014

*Guo et al., 2004*), but complete proper neural tube closure (*Guo et al., 2004*; *Wang et al., 2006*). Cell geometries and division orientations were unchanged in these mutants, further suggesting open neural tube defects impact epidermal division planes independently of PCP function (*Figure 5—figure supplement 1*).

To further test this hypothesis, we asked whether epidermal-specific expression of Vangl2-GFP could rescue the cell geometry and division orientation defects of Vangl2 mutants. Expression of Vangl2-GFP under the K14-promoter rescued PCP protein asymmetry (*Figure 6A*) and hair follicle orientations in the epidermis (*Devenport et al., 2011*), but not the neural tube closure defect. Similar to $Vangl2^{\Delta Tm/Lp}$ mutants, basal epidermal cells of $K14\text{-}Vangl2\text{-}GFP$; $Vangl2^{\Delta Tm/Lp}$ embryos had decreased cross-sectional areas, were taller than $K14\text{-}Vangl2\text{-}GFP$; $Vangl2^{WT}$ control littermates, and packed more densely into the basal layer (*Figure 6B–D*). These embryos also displayed a greater proportion of oblique and perpendicular division orientations than control embryos (*Figure 6E,F*). The occurrence of these cell shape and division orientation defects, even in the presence of functioning epidermal PCP, again suggests that they are a secondary consequence of the open neural tube phenotype, and not directly influenced by planar cell polarity. We propose that failure of the neural folds to stretch the surface ectoderm over the midline in PCP mutants leads to crowding and narrowing of basal cells in the flanking epidermis, biasing divisions planes toward oblique and perpendicular orientations.

## Spatial differences in epithelial packing and shape correlate with cell division orientations and the timing of stratification

To investigate whether cell shape and density regulate division orientations in the context of normal skin development, we examined an earlier stage of epidermal development, E13.5, when basal cell geometries naturally vary in a spatially defined pattern (*Aw et al., 2016*). Along the midline, basal cells are stretched over the spinal cord whereas laterally, cell shapes are more isometric. We quantified cell morphologies in three distinct regions of the skin and observed large differences in cell density and geometry along the mediolateral axis (*Figure 7A*). Basal cells along the midline were highly flattened compared to cells at intermediate and lateral positions (average cell height = 4.6 μm at the midline vs 6.9 μm laterally; *Figure 7B*). This reduction in cell height at the midline was accompanied by a large increase in cross-sectional area (average XY surface area = 363.5 μm$^2$ in the midline vs 164.9 μm$^2$ and 157.9 μm$^2$ and at intermediate and lateral zones, respectively) and a decrease in cell density (*Figure 7C,D*). Interestingly, although cells at lateral and intermediate positions had similar heights and surface areas, those at intermediate positions were significantly more elongated (*Figure 7E*; *Aw et al., 2016*), suggesting the curvature of the neighboring midline exerts mediolateral tension upon cells in this intermediate space.

We next examined cell division orientations in each region and found that division planes correlated with spatial differences in cell geometry. Although oblique and perpendicular divisions are relatively rare at E13.5, the majority were observed within lateral regions of the embryo where cells are taller and more crowded compared to medial regions (*Figure 7F–H*). In the intermediate space, we observed an increased frequency of planar divisions relative to the lateral region, likely as a result of the increased cell elongation in this space. Along the midline, where cells are flatter and have greater planar surface areas, divisions were almost exclusively planar (94% of divisions) (*Figure 7I–L*). These spatial differences in cell geometry and division plane were not observed between lateral and intermediate zones of $Vangl2^{Lp/\Delta Tm}$ mutants whose skin fails to close over the midline (*Figure 7—figure supplement 1*). Thus, coverage of the epidermis over the spinal cord induces a gradient of cell density and elongation along the mediolateral axis, which in turn regulates cell division planes across the epidermis.

## Exogenous stretching of skin explants increases the frequency of planar cell divisions

Our results thus far indicate that basal epidermal cells can orient their division planes to respond to changes in packing and geometry. The neural tube defect of PCP mutants offered a genetic tool with which to experimentally induce cell crowding and increase oblique and perpendicular division orientations. Therefore, we reasoned that mechanical manipulations that stretch the epidermis should bias cell divisions toward planar orientations. To test this, we cultured embryonic skin

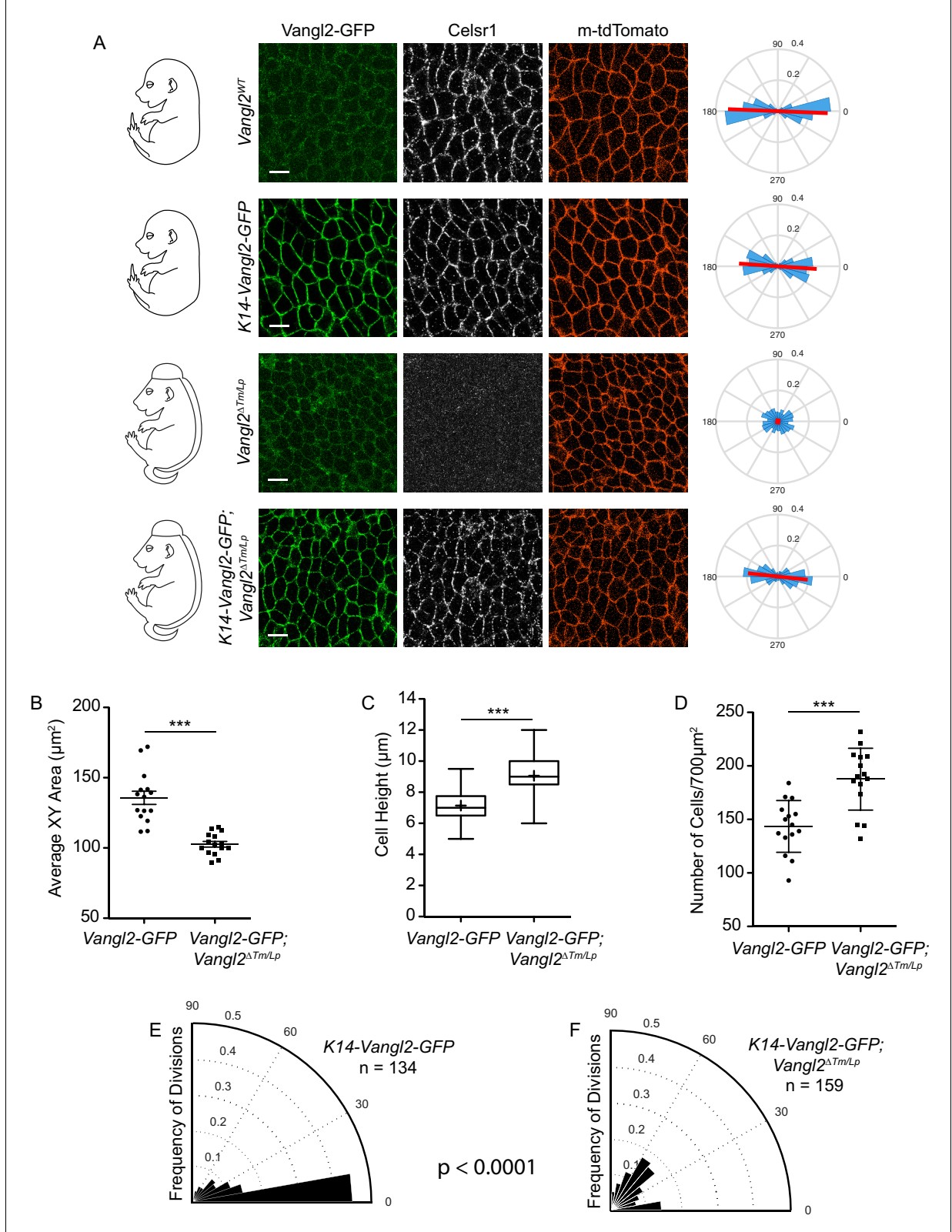

**Figure 6.** Cell shape and division orientation defects in Vangl2 mutants are indirect, and likely a consequence of failed neural tube closure. (A) Representative images of the basal layer of *Vangl2^WT* and *Vangl2^ΔTm/Lp* embryos with or without *K14-Vangl2-GFP*. Green = 488 channel, Greyscale = Celsr1, Red = membrane tdTomato. Rose plots quantify Celsr1 polarity in the basal layer. Red line indicates the average magnitude and direction of polarity. Rose plots were generated from three images per genotype. All scale bars = 10 μm. (B) Quantification of cell surface areas. Each

*Figure 6 continued on next page*

*Figure 6 continued*

dot represents average surface area of all cells in a single field of view. n = 15 fields of view across three embryos per genotype. Bars are mean with SEM. Unpaired two-tailed t-test, p<0.0001. (C) Quantification of cell height along apical-basal axis. n = 300 cells per genotype. Whiskers indicate minimum and maximum values, + indicates the mean. Unpaired two-tailed t-test, p<0.0001. (D) Quantification of cell density, as number of cells per field of view (700 µm$^2$). Each dot represents a single field of view. n = 15 fields of view across three embryos per genotype. Bars are mean with SD. Unpaired two-tailed t-test, p<0.0001. (E) Division orientations in *Vangl2$^{WT}$; K14-Vangl2-GFP* embryos. n = 134 divisions, from 45 images across three embryos. (F) Division orientations in *Vangl2$^{\Delta Tm/Lp}$; K14-Vangl2-GFP* embryos. n = 159 divisions, from 45 images across three embryos. Modified Kuiper Test, p=1.6201e-19.

DOI: https://doi.org/10.7554/eLife.47102.015

explants on flexible substrates and applied uniaxial stretch across the tissue (*Aw et al., 2016*). To determine whether planar division orientations could be restored in E14.5 *Vangl2$^{\Delta Tm/Lp}$* skin explants, we stretched along the medial-lateral axis to mimic the tension experienced by the epidermis when the neural tube has closed. Stretching E14.5 *Vangl2$^{\Delta Tm/Lp}$* skin explants significantly increased the planar elongation of basal cells (average elongation = 30.79 in stretched explants, vs 20.30 in control explants) and reduced cell heights as compared to unstretched controls (*Figure 8A–C*). Additionally, while basal cells of *Vangl2$^{\Delta Tm/Lp}$* control explants underwent primarily oblique and perpendicular divisions, stretched *Vangl2$^{\Delta Tm/Lp}$* explants exhibited an increased proportion of planar division orientations (62.5% planar divisions in stretched explants vs 27.1% in control) (*Figure 8D,E*).

We next investigated the effect of stretch on wildtype explants to test whether basal cells can dynamically reorient division planes in response to changes in cell geometry. Instead of stretching E14.5 explants when divisions already occur in primarily planar orientations, we utilized E15.5 explants, which undergo increased oblique and perpendicular divisions relative to earlier developmental stages. When E15.5 explants were stretched over the course of 1 hr, basal cells became elongated along the axis of strain and divisions were shifted toward planar division orientations compared to unstretched controls (*Figure 8—figure supplement 1*). The ability to rescue the division orientation defect of *Vangl2$^{\Delta Tm/Lp}$* embryos and promote planar divisions in wildtype explants through uniaxial stretch demonstrates that basal cells are able to integrate tissue dynamics with cell division orientation to respond to and influence tissue architecture.

## Discussion

By live imaging of the developing epidermis, we have shown that the orientation of cell division within the skin's basal progenitor layer is closely linked to the positional fate of daughter cells. Basal cells dividing with telophase angles greater than 45 degrees of the basement membrane will have asymmetric fates, whereas division angles less than 45 degrees will generate daughters with symmetric fates. Although this has been assumed to be true based on division planes in fixed tissue sections, these data show directly that during early phases of epidermal stratification, the angle between telophase daughter cells is a strong indicator of the final positions and fates of the two daughters. We also found, through investigation of the hyperstratification defect of a well-studied PCP mutant, that during early stratification stages, the orientation of cell division is influenced by cell shape and packing rather than cortical PCP cues or LGN localization. Basal cells that are taller along the apical-basal axis are biased towards perpendicular divisions, while flatter and wider cells tend to divide parallel to the epithelial plane. Crowding within the epithelium, a consequence that is likely due to a failure of the skin to stretch and cover the midline in PCP mutants, is associated with increased asymmetric divisions, while stretching the epithelium promotes planar divisions and expansion of the progenitor layer. These findings show that embryonic basal epidermal cells are flexible to adjust their divisions, at the level of spindle orientation, in response to changes in the tissue environment. We propose that under normal developmental conditions, this mechanism enables expansion of the progenitor layer to accommodate embryo growth while also allowing for the generation of new layers when the basal layer has reached sufficient density.

Based on these observations, we propose an updated model for epidermal stratification. The epidermis begins as a single layer of progenitor cells that divide almost exclusively within the plane of the tissue, until increases in cell density and height:width ratio promote stratification through oblique/perpendicular divisions. Epithelial stretch or tension, such as that exerted on the skin by embryo

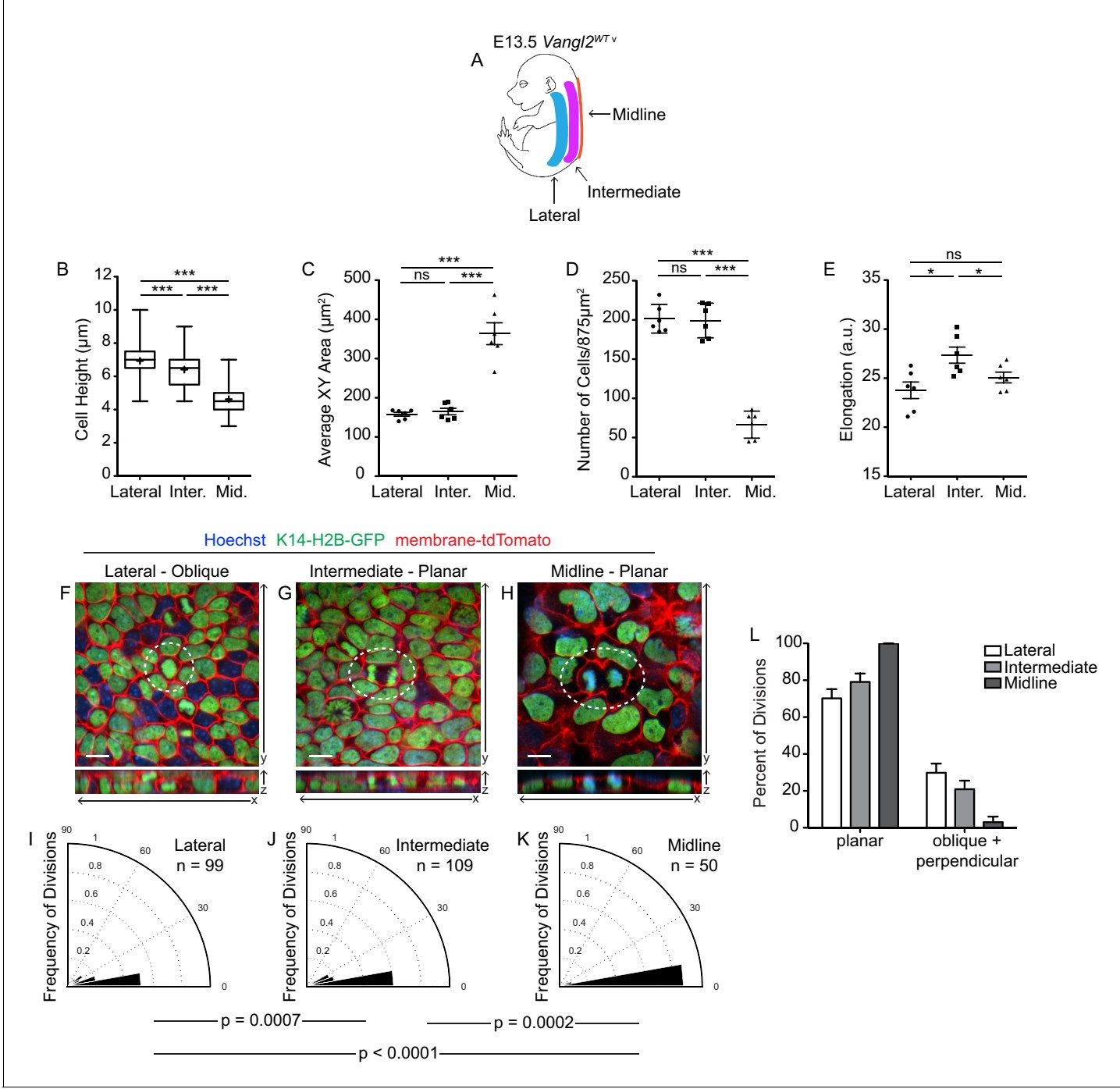

**Figure 7.** Spatial differences in cell geometries are associated with regional changes in division orientations. (**A**) Schematic of an E13.5 *Vangl2*<sup>WT</sup> embryo, depicting the lateral, intermediate, and midline regions. (**B**) Quantification of cell height along apical-basal axis. n = 300 cells per genotype. Whiskers indicate minimum and maximum values, + indicates the mean. Lateral vs Intermediate, unpaired two-tailed t-test, p<0.0001; Lateral vs Midline, unpaired two-tailed t-test, p<0.0001; Intermediate vs Midline, unpaired two-tailed t-test, p<0.0001. (**C**) Quantification of basal cross-sectional areas. Each dot represents average surface area of all cells in a single field of view. Two images per region were analyzed, from each of three embryos, for a total of 6 measurements per region. Bars are mean with SEM. Lateral vs Intermediate, unpaired two-tailed t-test, p=0.4872; Lateral vs Midline, unpaired two-tailed t-test, p<0.0001; Intermediate vs Midline, unpaired two-tailed t-test, p<0.0001. (**D**) Quantification of cell elongations. Each dot represents the average elongation value of all cells in a field of view. Two images per region were analyzed, from each of three embryos, for a total of 6 measurements per region. Bars represent mean with SEM. Lateral vs Intermediate, unpaired two-tailed t-test, p=0.0125; Lateral vs Midline, unpaired two-tailed t-test, p=0.2256; Intermediate vs Midline, unpaired two-tailed t-test, p=0.0432. (**E**) Quantification of cell density, as number of cells per image (875 μm²). Each dot represents a single field of view. Two images per region were analyzed, from each of three embryos, for a total of 6 measurements

*Figure 7 continued on next page*

*Figure 7 continued*
per region. Bars represent mean with SD. Lateral vs Intermediate, unpaired two-tailed t-test, p=0.8567; Lateral vs Midline, unpaired two-tailed t-test, p<0.0001; Intermediate vs Midline, unpaired two-tailed t-test, p<0.0001. (F–H) Representative images of cell divisions in the lateral, intermediate, and midline regions in E13.5 *Vangl2^{WT}; K14-H2B-GFP* embryos. Hoechst and membrane-tdTomato mark the nuclei and cell membranes, respectively. (I–K) Division orientations in the lateral, intermediate, and midline regions. Lateral: n = 99 divisions pooled from three embryos; Intermediate: n = 109 divisions pooled from three embryos; Midline: n = 50 divisions pooled from three embryos. Modified Kuiper's Test: lateral vs intermediate, p=7.2638e-04; lateral vs midline, p=9.8559e-06; intermediate vs midline, p=1..5851e-04. (L) Distribution of planar vs oblique + perpendicular division orientations in the lateral, intermediate, and midline regions.
DOI: https://doi.org/10.7554/eLife.47102.016
The following figure supplement is available for figure 7:

**Figure supplement 1.** Spatial differences in cell geometry and division orientation are lost in Vangl2 mutant embryos.
DOI: https://doi.org/10.7554/eLife.47102.017

growth or morphogenetic changes like neural tube closure, delays stratification, while crowding of the basal layer promotes its occurrence. These cues of stretch and crowding provide an effective way to balance basal cell division orientations and ensure proper tissue structure early in epidermal development. Notably, other studies using Cre recombinase-based lineage tracing or live imaging without a nuclear marker have suggested that stratification at this early stage is driven primarily by delamination (*Williams et al., 2014*; *Miroshnikova et al., 2018*; *Wickström and Niessen, 2018*). By contrast, we did not observe delamination events throughout the analysis of our live imaging data. The reasons for these differences are not entirely clear, but one possibility is that lentiviral delivery of CreER might promote delamination of less fit basal cells into the suprabasal layer. Additionally,

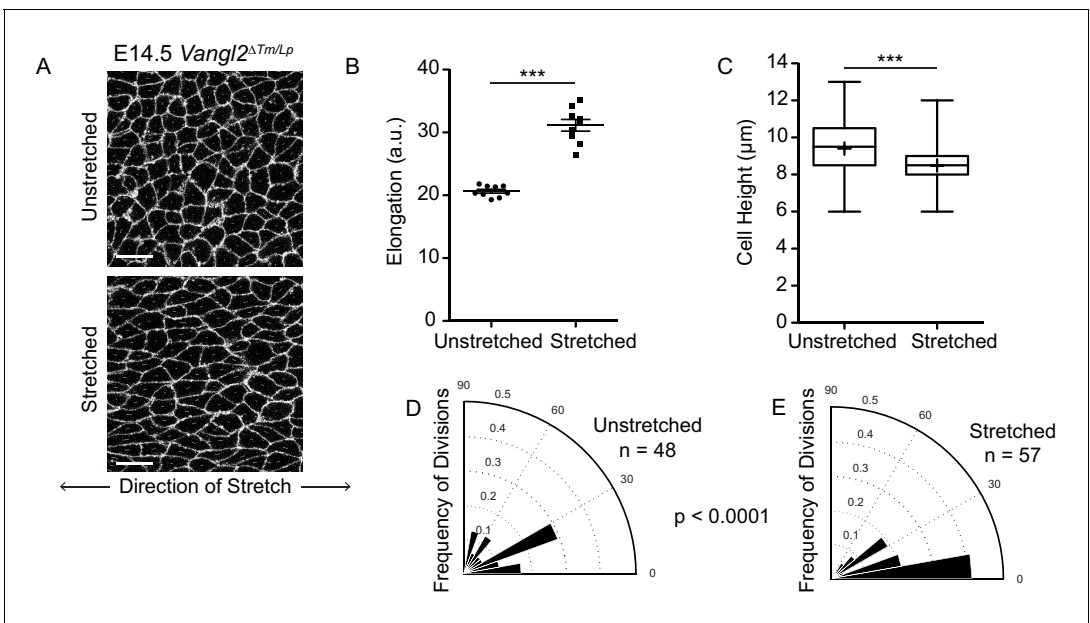

**Figure 8.** Basal cell divisions shift towards planar orientations under exogenously applied stretch. (A) Representative images of the basal layer of E14.5 *Vangl2^{ΔTm/Lp}* unstretched and stretched skin explants. White, membrane-tdTomato. Scale bars = 10 μm. (B) Quantification of cell elongation in unstretched vs stretched skins. Each dot represents the average elongation value of all cells in a field of view. n = three images from each of three separate experiments. Bars are mean with SEM. Unpaired two-tailed t-test, p<0.0001. (C) Quantification of cell height along apical-basal axis. n = 450 cells per genotype. Whiskers indicate minimum and maximum values, + indicates the mean. Unpaired two-tailed t-test, p<0.0001. (D) Division orientations in unstretched E14.5 *Vangl2^{ΔTm/Lp}* explants. n = 48 divisions pooled from three explants. (E) Division orientations in stretched E14.5 *Vangl2^{ΔTm/Lp}* embryos. n = 57 divisions pooled from three explants. Modified Kuiper's Test, p=3.9430e-06.
DOI: https://doi.org/10.7554/eLife.47102.018
The following figure supplement is available for figure 8:

**Figure supplement 1.** Exogenously applied stretch shifts E15.5 basal cell divisions toward planar orientations.
DOI: https://doi.org/10.7554/eLife.47102.019

we noted that cells undergoing perpendicular divisions can appear as if they are delaminating when the focal plane aligns with the plane of cleavage (see *Figure 4—video 2*; *Miroshnikova et al., 2018*). Inclusion of a nuclear marker and Z-dimensional reconstructions are required to resolve such ambiguities.

Late in embryonic development, there is a developmental shift toward generating more skin layers, as forming an effective barrier from the exterior environment becomes imperative. To facilitate this change in tissue architecture, apical mInsc polarization promotes perpendicular spindle orientations and formation of a stratified barrier (*Lechler and Fuchs, 2005*; *Poulson and Lechler, 2010*; *Williams et al., 2011*; *Williams et al., 2014*). During homeostasis of the adult epidermis, yet another shift occurs in the balance between division and stratification. Division angles are once again primarily planar and delamination, rather than asymmetric division, becomes the mechanism by which cells are able to replenish the skin's stratified layers (*Clayton et al., 2007*; *Mesa et al., 2018*; *Rompolas et al., 2016*; *Smart, 1970*). During adult homeostasis, differentiation triggers neighboring basal stem cells to divide. Thus, similar to what we've observed in early epidermal stages, homeostasis is influenced by the needs of the tissue, as sensed through local tissue architecture (*Mesa et al., 2018*).

The different strategies that basal cells employ to orient divisions as the epidermis develops raises many new questions that warrant further investigation. Firstly, it is unclear what regulates the developmental switches in division modes. For example, the switch to mInsc/LGN/NuMA-mediated perpendicular divisions coincides with an increase in mInsc polarization (*Lechler and Fuchs, 2005*; *Poulson and Lechler, 2010*; *Williams et al., 2011*; *Williams et al., 2014*), but how this change in polarization is regulated is unknown. Additionally, we do not know precisely when or how the switch back to planar divisions during homeostasis occurs. Secondly, the cortical cues that orient planar spindle alignment, at any developmental stage, remain to be identified. In single-layered epithelia, such as the *Drosophila* follicular epithelium and the chick neuroepithelium, planar spindle orientation depends on the LGN/Pins-NuMA/Mud complex, which anchors astral microtubules to the lateral cell cortex through the polarity factor Dlg (*Bellaïche et al., 2001b*; *Bergstralh et al., 2013*; *Nakajima et al., 2013*; *Saadaoui et al., 2014*). Murine epidermal progenitors require LGN-NuMA for perpendicular spindle orientation during the latter phase of stratification, but these proteins are dispensable for planar spindle alignment (*Williams et al., 2014*). Moreover, although core PCP components orient parallel divisions in several other contexts (*Bellaïche et al., 2004*; *Bellaïche et al., 2001a*; *Ciruna et al., 2006*; *Fischer et al., 2006*; *Gong et al., 2004*; *Ségalen et al., 2010*), the core PCP component Vangl2 is not required for parallel, symmetric divisions in the skin. Conditional ablation of β1 integrin or α-catenin leads to randomization of division planes in the basal layer (*Lechler and Fuchs, 2005*) suggesting planar spindle alignment requires both basal and lateral adhesion proteins. However, cell polarity might be generally compromised in these mutants. Thus, the cortical cues that pull the spindle in a planar orientation still need to be identified, and it is unclear whether such factors would be localized to the lateral or basal cortex. Finally, the mechanisms that connect spindle orientation to cell geometry in the early stages of stratification remain unknown.

Although the propensity for a cell to divide along its longest interphase axis, Hertwig's rule, has been known for over 100 years, it is not entirely intuitive how the mitotic spindle, which assembles and anchors only after the cell has rounded, orients relative to the cell's geometry when it was in interphase. Evidence that cells retain a 'cortical memory' of their interphase shape has been shown in the pupal notum epithelium of *Drosophila* and single cells in culture. In planar divisions of the *Drosophila* notum, NuMA localizes to tricellular junctions, where it captures astral microtubules to dictate division orientation. Because tricellular junction position is preserved from interphase, their position serves as a memory of interphase geometry when the cell rounds up during mitosis (*Bosveld et al., 2016*). When HeLa cells are plated on fibronectin micropatterns, retraction fibers left behind during mitotic rounding present spatial cortical cues that can bias the localization of subcortical molecular factors, which in turn influences mitotic spindle orientation (*Fink et al., 2011*; *Théry et al., 2005*). Ablation of these fibers or adjustment of their distribution through pattern variation affects spindle positioning (*Fink et al., 2011*). In the murine epidermis, however, the mechanisms responsible for communicating cell shape and/or tension information to the mitotic spindle are unknown. Whether the same principles by which cell geometry orients divisions in two dimensions apply to the three dimensional division planes of basal cells is also unclear. Because each cell is dividing in the context of the surrounding tissue, perhaps neighboring cells exert tension on the

dividing cell as shared junctions are distorted due to cell shape changes that occur during mitosis. For example, cells that are elongated within the plane of the epidermis would have the greatest tension exerted upon them from lateral neighbors as they progress from an elongated to a rounded shape, and thus would be more likely to divide in a planar orientation. Conversely, cells that have more isometric cross-sectional areas and are elongated along their apical-basal axes may lack these planar forces. Alternatively, there may be a cell intrinsic system for interphase shape memory that guides the spindle to align along its interphase 3D long axis. Ultimately, deciphering the processes by which basal epidermal cells connect local tissue mechanics to cell division orientations will illuminate a new method of cell fate control.

# Materials and methods

## Key resources table

| Reagent type (species) or resource | Designation | Source or reference | Identifiers | Additional information |
|---|---|---|---|---|
| Strain, strain background (*M. Musculus*) | CD1 | Charles River | Crl:CD1(ICR); Strain 022 | |
| | C56BL/6 | The Jackson Laboratory | C57BL/6J; Stock No: 000664 | |
| | Vangl2 (Lp) | *Kibar et al., 2001* | MGI: 1857642 | Elaine Fuchs |
| | Vangl2 (dTM) | *Copley et al., 2013* | MGI: 5551989 | Michael Deans |
| | K14-H2B-GFP | *Tumbar et al., 2004* | | Elaine Fuchs |
| | K14-Cre | *Vasioukhin et al., 1999* | | Elaine Fuchs |
| | Rosa26-mTmG | *Muzumdar et al., 2007* | MGI: 3716464 | Liqun Luo |
| | Vangl1 flox/flox | *Wang et al., 2016* | MGI: 5440498 | Jeremy Nathans |
| | Vangl2 flox/flox | *Copley et al., 2013* | MGI: 5551989 | Michael Deans |
| | K14-Vangl2-GFP | *Devenport et al., 2011* | | Elaine Fuchs |
| | Frizzled-6 KO | *Guo et al., 2004* | MGI: 3050103 | Saori Haigo and Jeremy Reiter |
| Antibody | anti-Involucrin (rabbit polyclonal) | Covance | | (1:500), Elaine Fuchs |
| | anti-Celsr1 (guinea pig polyclonal) | D Devenport | | (1:1000) |
| | anti-E-Cadherin (rat monoclonal) | Masatoshi Takeichi | | (1:25) purified in lab from ECCd2 clonal hybridomas |
| | anti-E-Cadherin (rat monoclonal) | Thermo Fisher | MA1-25160 | (1:2000) |
| | anti-LGN (guinea pig polyclonal) | Scott Williams | | (1:500) |
| | anti-LGN (rabbit polyclonal) | Scott Williams | | (1:3000) |
| | anti-Vangl2 (rat monoclonal) | Millipore | Cat: MABN750 | (1:100) |
| | Alexa Fluor-488; −555; −647 secondary antibodies | Invitrogen | | (1:2000) |

*Continued on next page*

*Continued*

| Reagent type (species) or resource | Designation | Source or reference | Identifiers | Additional information |
|---|---|---|---|---|
| | Alexa Fluor-488; −555; −647 secondary antibodies | Jackson Immuno-Research | | (1:2000) |
| Software, algorithm | Packing Analyzer | *Aigouy et al., 2010* | | Benoit Aiguoy |

## Mouse lines and breeding

All procedures involving animals were approved by Princeton University's Institutional Animal Care and Use Committee (IACUC). Mice were housed in an AAALAC-accredited facility in accordance with the Guide for the Care and Use of Laboratory Animals. This study was compliant with all relevant ethical regulations regarding animal research. E13.5-E16.5 embryos from C57BL/6 backgrounds were used unless otherwise indicated. Both sexes were used, as sex was not determined in embryos. Mouse lines harboring the spontaneous *Looptail* mutation (*Vangl2^{Lp}*) lines generated extremely high frequencies of hermaphrodites, severely limiting the availability of female animals. Therefore, females of the genotype *Vangl2^{Lp}* or *Vangl2^{ΔTm}*, a targeted deletion of the second transmembrane domain, were used for experiments (see Key Resources Table).

## Whole mount immunostaining

For immunostaining, embryos were dissected in PBS and fixed in 4% paraformaldehyde. E13.5 and E14.5 embryos and embryonic explants were fixed at room temperature for 1 hr, and E15.5 and E16.5 embryos and embryonic explants were fixed at room temperature for 1.5 hr. Skins stained with anti-LGN antibodies were processed as follows: skins were dissected from fixed embryos and blocked overnight at 4°C in 2% fish skin gelatin, 1% bovine serum albumin, 2% normal donkey serum, and/or 2% normal goat serum in PBT (1X PBS with 0.2% or 0.3% Triton X-100). Skins were incubated in primary antibodies diluted in block for 24 hr at 4°C. Skins were washed with 0.2–0.3% PBT overnight, incubated with secondary antibodies for 24 hr at 4°C followed by Hoechst. All other skins were blocked for 1 hr at RT in 1% BSA, 2% NDS and/or 2% NGS then incubated in primary antibodies diluted in block overnight at 4°C. After washing with 0.2–0.3% PBT, skins were incubated in secondary antibodies followed by Hoechst. Samples were mounted in either glycerol-based antifade non-curing medium or Vectashield Antifade mounting medium. The following primary antibodies were used: rabbit anti-involucrin (1:500, D. Devenport), guinea pig anti-Celsr1 (1:1000, D. Devenport), rat anti-E-Cadherin (1:25, purified in lab from ECCD2 clonal hybridomas, obtained from Masatoshi Takeichi), rat anti-E-Cadherin (1:2000, DECMA-1, Thermo Fisher, Cat: MA1-25160), guinea pig anti-LGN (1:500, gift from Scott Williams), rabbit anti-LGN (1:3000, gift from Scott Williams), rat anti-Vangl2 (1:100, Millipore, Cat: MABN750). Alexa Fluor-488,−555, −647 secondary antibodies were used at 1:2000. Hoechst (Invitrogen, Cat: H1399) was used at 1:1000 (1 mg/mL). Images were acquired on a Nikon Ti-E Spinning Disc or A1 scanning confocal microscope controlled by NIS Elements software using a Plan Fluor 40X/1.3NA, Plan Apo 60X/1.4NA, or Plan Apo 100X/1.45 oil immersion objective. NIS elements software and Photoshop were used for image processing.

## Live imaging

A portion of live imaging for *Figure 1* was done as follows: E14.5 flank skin explants were dissected in PBS and mounted on bottom side of Sarstedt 35 mm Lumox membrane dish (Cat# 94.6077.331) and secured in position using 13 mm diameter, 8 mm Nucleopore polycarbonate membrane (Fisher Scientific Cat# 09-300-57) overlaid with matrigel (Fisher Cat# CB-40230A). This dish was then mounted inside a Sarstedt standard TC dish (Cat# 83.3900) filled with F-Media with 10% fetal bovine serum. This setup was necessary to enable imaging from the epidermal side of the explant using an upright microscope. Time-lapse imaging was undertaken using an upright Prairie Ultima multiphoton microscope equipped with 40x plan apochromat, 1.0NA immersion objective with incubation chamber to maintain 37°C and 5% CO2. 1024 × 1024 pixel images were collected in Z-stacks spaced at

1.0 mm apart. 3D Z-stacks were collected at 10 min intervals for 24 hr. Volocity software was used for movie processing.

For all the other movies: E14.5 dorsal flank skin explants were dissected in PBS and transferred to a 1% agarose gel with F-media containing 10% fetal bovine serum. Explants were sandwiched between the gel on the dermal side and a 35 mm Lumox membrane dish (Sarstedt) on the epidermal side. Z-stacks with 0.5 or one micron step sizes were acquired at 10 min intervals for 8–18 hr. Images were acquired using a Nikon Ti-E Spinning Disc with a Plan Fluor 40X/1.3NA oil objective. Explants were cultured in a humid imaging chamber at 37˚C with 5% $CO_2$ during the course of imaging. In all live imaging experiments, cells close to the edge of the explants were not imaged to avoid differences in division orientations that could be induced by a wound healing response. NIS Elements software was used for movie processing.

## Stretching experiments

A custom-designed stretch chamber (*Aw et al., 2016*) was constructed to culture and uniaxially stretch skin explants on a flexible 7 × 3 cm PDMS membrane (SSP-M823-005, Specialty Silicone Products, Inc). The PDMS membranes were exposed to UV light for 15–20 min then coated with fibronectin. E14.5 or E15.5 dorsal flank skin explants were dissected in PBS and positioned dermis side down on the membranes and submerged in E-media containing 1.5 mM calcium. E14.5 explants were cultured with 10% mouse serum, isolated in the lab. The explants were allowed to rest for 3 hr before stretching. One half of each explant was cultured in a similar chamber with no applied stretch, as a control. Experiments were performed by stretching the membrane 4 mm over the course of 1 hr. Skin explants were kept on the PDMS membranes for 15 min following stretching, then fixed at room temperate for 1 hr, then removed from the PDMS membrane and mounted in Vectashield mounting medium for imaging. Basal cell elongation was quantified using Packing Analyzer V2 software, which determines the direction and magnitude of cell elongation as previously described (*Aw et al., 2016*).

## Quantification of division angle, skin thickness, cell height, cell surface area

Division angle in both live and fixed imaging was determined by drawing a line connecting the two daughter nuclei during telophase or cytokinesis (using either Volocity or NIS Elements software). Skin thickness was quantified using fixed sagittal tissue sections or whole mount skins and by measuring the area of skin in an image, normalized to length. For whole mount images, area and length were measured in YZ views. Cell heights were measured using membrane signal (either E-Cadherin or membrane-tdTomato, held consistent within each experiment) and XZ views of images with either 0.2 μm or 0.5 μm Z-step sizes in NIS Elements software to determine the distance from the apical surface of the cell to the basement membrane. Cell surface area was calculated using Packing Analyzer V2 software. Images were imported into the software and cells were segmented using a membrane signal, with any necessary hand corrections. Surface areas were calculated using the software's watershed formula (*Aw et al., 2016*).

## Statistics

Circular statistics: to compare rose plots of division angles, p values were calculated from the nonparametric Kuiper's test to compare distributions along a circle. This test was modified to account for data limited to 0–90˚ range, and p values were calculated using Matlab. All other statistical analyses were performed in Prism.

## Acknowledgements

We thank Scott Williams for providing LGN antibodies, Benoit Aiguoy for Packing Analyzer, Benjamin Bratton for assistance with circular statistics, and Gary Laevsky and the Nikon Center for Excellence for technical assistance with imaging. We are grateful to Gertrud Schüpbach, Mark Rose, Rebecca Burdine, and Devenport lab members, especially Maureen Cetera, for helpful discussions and feedback. Work was supported by NIH/NIAMS grant number R01AR068320 and NIH/NIGMS grant number T32GM007388.

## Additional information

### Funding

| Funder | Grant reference number | Author |
|---|---|---|
| National Institutes of Health | R01AR068320 | Danelle Devenport |
| National Institutes of Health | T32GM007388 | Kimberly Box |
| National Institutes of Health | AR8472471 | Kimberly Box<br>Danelle Devenport |

The funders had no role in study design, data collection and interpretation, or the decision to submit the work for publication.

### Author contributions

Kimberly Box, Conceptualization, Data curation, Formal analysis, Investigation, Methodology, Writing—original draft, Writing—review and editing, Designed and performed the experiments and analyzed the data; Bradley W Joyce, Resources, Data curation, Methodology, Contributed a portion of live imaging for Figure 1; Danelle Devenport, Conceptualization, Supervision, Funding acquisition, Writing—original draft, Writing—review and editing

### Author ORCIDs

Danelle Devenport (iD) https://orcid.org/0000-0002-5464-259X

### Ethics

Animal experimentation: All procedures involving animals were approved by Princeton University's Institutional Animal Care and Use Committee (IACUC) under protocol #1867. Mice were housed in an AALAC-accredited facility in accordance with the Guide for the Care and Use of Laboratory Animals. This study was compliant with all relevant ethical regulations regarding animal research.

### Decision letter and Author response

Decision letter https://doi.org/10.7554/eLife.47102.023
Author response https://doi.org/10.7554/eLife.47102.024

## Additional files

### Supplementary files

• Source code 1. Matlab codes for data analysis.
DOI: https://doi.org/10.7554/eLife.47102.020

• Transparent reporting form
DOI: https://doi.org/10.7554/eLife.47102.021

### Data availability

All measured data are reported as their full distributions in the figures and supplements. Source data files with individual measurements are provided for all figures. Matlab codes for data analysis are provided in Source code 1.

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
