## [Decision Letter]

Thank you for submitting your article "Epithelial geometry regulates spindle orientation and progenitor fate during formation of the mammalian epidermis" for consideration by *eLife*. Your article has been reviewed by Marianne Bronner as the Senior Editor, a Reviewing Editor, and two reviewers. The following individuals involved in review of your submission have agreed to reveal their identity: Yukiko M Yamashita (Reviewer #1); Yohanns Bellaïche (Reviewer #3).

The reviewers have discussed the reviews with one another and the Reviewing Editor has drafted this decision to help you prepare a revised submission.

Summary:

This study by Box et al., studies the role of PCP protein Vangl2 in division orientation of basal progenitor cells (epidermal stem cells) in mouse. Overall, the conclusions (summarized below) are supported well, and the study represents important description on spindle orientation and tissue morphogenesis, including the descriptions that correct notions that might have been assumed based on previous studies. This study is the first to provide much clear-cut correlation between the spindle orientation and the fate of daughter cells. They also made an important observation that contrasts to the previous studies that suggested delamination of basal progenitor cells as a mechanism of differentiation.

Essential revisions:

-Comparing Figure 4I and 4J, one interesting analysis may be whether wt vs. Vangl2 mutant cells have similar height-width ratio in deciding which direction they divide. Namely, Vangl2 mutation only affects height-width ratio, and all spindle orientation phenotype is purely explained by changed height-width ratio. Alternatively, Vangl2 might possibly affecting additional aspect(s) of spindle orientation other than height-width ratio? By eye-balling Figure 4I and 4J, it looks like there is differences between wt vs. mutant for the same category of spindle orientation.

-Can they explain why the previous studies proposed delamination? I am wondering the studies relied on fixed samples? If so, is it possible that maybe some daughter cells are coming back to adhere basal layer? (opposite of delamination?)

Subsection “Division plane is a strong predictor of the positional fate of daughter cells”: The following statement might be incorrect: "*Vangl2Lp/155 Lp* embryos, however, the relationship between division orientation and final position was shifted toward asymmetric. Only 78% of planar divisions, and 21% of oblique divisions resulted in symmetric positional fates (Figure 2B,C).". The distribution shown in the Figure 2A seems similar. The distribution difference in the Figure 2B could be due to the binning in planar, oblique, orthogonal divisions. To strengthen this statement, the authors should test whether the angular distributions of the different bins are identical.

Results section/Discussion section: In the Results section or the Discussion section, it would be relevant to discuss the difference between this study and the ones reporting stratification by delamination.

Subsection “Division plane is a strong predictor of the positional fate of daughter cells”: The authors should clarify what is novel in the analysis of the epithelium thickness in Vangl2 mutant embryos.

Figure 3: "LGN Mislocalized" sounds inappropriate. An absence of correlation between LGN localization and a specific orientation does not indicate that LGN is mis-localised.

Subsection “Early in epidermal development, LGN localization does not correlate with planar spindle positioning”: This sentence: "However, the occurrence of apical LGN (47.7% of divisions) was not sufficiently increased compared to *Vangl2^WT^*(41.1% of divisions) to explain the altered division orientations (Figure 3C, D)" is awkward since the frequency is decreased. The subsequent sentence is incorrect, the conclusion should be made on Vangl2 and not LGN.

Subsection “Cell division orientation is correlated with basal cell geometry”: "but this has been shown mainly in cells dividing within a two-dimensional plane (Bosveld et al., 2016; Thery and Bornens, 2006; Thery et al., 2005; Wyatt et al., 2015)." This is an overstatement. Xiong et al., (2014) have analysed the role of 3D cell shape. This work needs to be cited and discussed.

Subsection “Cell division orientation is correlated with basal cell geometry”: "Basal cells in *Vangl2^Lp/Lp^*embryos were packed more densely into the basal layer compared to wildtype (258 cells vs 215 cells per 1200μm2 field of view) (Figure 4E), but this increase in cell density was not due to elevated proliferation (Figure 4F).". Small difference in the proliferation rate can have a substantial impact on cell numbers at long time scale. Please reformulate.

Subsection “Cell division orientation is correlated with basal cell geometry”: "We also observed a modest but significant relationship between division plane and height:width aspect ratio along the apical-basal axis, wherein cells that divided in a planar orientation tended to have smaller height:width ratios (average = 0.8307) compared to cells that divided perpendicularly (average height:width ratios = 1.275) (Figure 4I). This relationship was also observed in Vangl2 mutant embryos (Figure 4H, J, Figure 4—video 2), where perpendicular divisions were associated with larger height:width ratios". The authors need to plot cell division orientation versus height: width ratio to substantiate this conclusion.

Subsection “Spatial differences in epithelial packing and shape correlate with cell division orientations and the timing of stratification”: Are the orientation distributions statistically different in the lateral, intermediate and midline regions?

Subsection “Spatial differences in epithelial packing and shape correlate with cell division orientations and the timing of stratification”: "Thus, coverage of the epidermis over the spinal cord induces a gradient of cell density and elongation along the mediolateral axis, which in turn regulates cell division planes and the timing of stratification." Please weaken this conclusion.

Subsection “Exogenous stretching of skin explants increases the frequency of planar cell divisions”: The 14.5 control and stretched conditions should be done in wt conditions. Accordingly, the conclusion at the end of the Results section is incorrect. The authors can only make this conclusion in a Vangl2 mutant conditions.

---

## [Author Response]

Summary:This study by Box et al., studies the role of PCP protein Vangl2 in division orientation of basal progenitor cells (epidermal stem cells) in mouse. Overall, the conclusions (summarized below) are supported well, and the study represents important description on spindle orientation and tissue morphogenesis, including the descriptions that correct notions that might have been assumed based on previous studies. This study is the first to provide much clear-cut correlation between the spindle orientation and the fate of daughter cells. They also made an important observation that contrasts to the previous studies that suggested delamination of basal progenitor cells as a mechanism of differentiation.

We thank the reviewers for their careful reading of our manuscript and their helpful suggestions. We have addressed each of their comments and concerns and provide a point-by-point response below.

Essential revisions:-Comparing Figure 4I and 4J, one interesting analysis may be whether wt vs. Vangl2 mutant cells have similar height-width ratio in deciding which direction they divide. Namely, Vangl2 mutation only affects height-width ratio, and all spindle orientation phenotype is purely explained by changed height-width ratio. Alternatively, Vangl2 might possibly affecting additional aspect(s) of spindle orientation other than height-width ratio? By eyeballing Figure 4I and 4J, it looks like there is differences between wt vs. mutant for the same category of spindle orientation.

For this analysis, we plotted the proportion of cells dividing in each directional category binned by their H/W ratios (shown in Author response image 1). It does appear that cells of the same H/W ratio are slightly more likely to divide in oblique and perpendicular orientations in the *Vangl2^Lp/Lp^*mutant, but this difference was not significant (Chi-squared test for bin 1.0-1.5 p=0.52; bin 0.5-1.0 p=0.35), but this could perhaps be due to the low sample size within each bin. Thus, I don’t think we can conclude with our data set whether Vangl2 affects the spindle by variables other than shape.

-Can they explain why the previous studies proposed delamination? I am wondering the studies relied on fixed samples? If so, is it possible that maybe some daughter cells are coming back to adhere basal layer? (opposite of delamination?)

In Williams et al., 2014, lineage tracing of basal cell clones was performed by injecting lentivirus expressing CreER into Brainbow transgenic lines. One, two and three cell clones were analyzed in fixed skin samples. 50% of labeled cells were found in the first suprabasal layer with no associated basal cell, and were scored as delamination events. It’s possible that the viral infection itself promotes delamination if it affects basal cell fitness. In Miroshnikova et al., 2018, some live imaging was performed on life-Act expressing embryos lacking a nuclear marker. To our eyes, the example provided resembles what we see when a cell divides in a perpendicular orientation when visualized with a membrane marker (Figure 4-video 2). These differences are discussed in more detail in the Discussion section of the manuscript.

Subsection “Division plane is a strong predictor of the positional fate of daughter cells”: The following statement might be incorrect: "Vangl2Lp/155 Lp embryos, however, the relationship between division orientation and final position was shifted toward asymmetric. Only 78% of planar divisions, and 21% of oblique divisions resulted in symmetric positional fates (Figure 2B,C).". The distribution shown in the Figure 2A seems similar. The distribution difference in the Figure 2B could be due to the binning in planar, oblique, orthogonal divisions. To strengthen this statement, the authors should test whether the angular distributions of the different bins are identical.

To test whether the way we binned division categories affected our conclusion (0-20=planar, 21-70=oblique, 71-90=perpendicular), we binned the data in a different way (in 30 degree bins) and found the bias toward asymmetric fates in *Vangl2^Lp/Lp^*embryos to be even greater, suggesting our results aren’t artifacts of how we binned division planes. We decided to keep the original binning of these data and preserve the original language used in the text. We also provide a comparison of the data binned in different ways in Author response image 2.

**Author response image 2. respfig2:** 

Results section/Discussion section: In the Results section or the Discussion section, it would be relevant to discuss the difference between this study and the ones reporting stratification by delamination.

The differences between our study and those reporting delamination are now discussed in more detail in the Discussion section of the manuscript.

Subsection “Division plane is a strong predictor of the positional fate of daughter cells”: The authors should clarify what is novel in the analysis of the epithelium thickness in Vangl2 mutant embryos.

Although we had previously noted that *Vangl2^Lp/Lp^* mutants had thicker skin at E18.5, and images where this difference is apparent can be found in Devenport et al., 2008, we never explicitly mentioned the phenotype in prior publications. Here we quantify the increase in skin thickness for the first time and characterize how and when it emerges over the course of skin development. The novelty of these findings is now explained more clearly in the introduction and in the Results section accompanying Figure 2.

Figure 3: "LGN Mislocalized" sounds inappropriate. An absence of correlation between LGN localization and a specific orientation does not indicate that LGN is mis-localised.

We now refer to this population as ‘Not correlated’.

Subsection “Early in epidermal development, LGN localization does not correlate with planar spindle positioning”: This sentence: "However, the occurrence of apical LGN (47.7% of divisions) was not sufficiently increased compared to Vangl2^WT^ (41.1% of divisions) to explain the altered division orientations (Figure 3C, D)" is awkward since the frequency is decreased. The subsequent sentence is incorrect, the conclusion should be made on Vangl and not LGN.

We have rephrased this result and conclusion as follows, “Although the occurrence of apical LGN in *Vangl2^Lp/Lp^* embryos (47.7% of divisions) was slightly increased compared to *Vangl2^WT^*(41.1% of divisions), this increase is not sufficient to explain the altered division orientations (Figure 3C, D), suggesting that Vangl2 does not act through LGN to orient divisions. Together, these data are consistent with a potential role for Vangl2 in an alternative mechanism that influences division planes during the early phase of epidermal stratification.”

Subsection “Cell division orientation is correlated with basal cell geometry”: "but this has been shown mainly in cells dividing within a two-dimensional plane (Bosveld et al., 2016; Thery and Bornens, 2006; Thery et al., 2005; Wyatt et al., 2015)." This is an overstatement. Xiong et al., (2014) have analysed the role of 3D cell shape. This work needs to be cited and discussed.

Although we had cited these two studies, we now discuss Xiong et al., 2014 and Chalmers et al., 2003 more explicitly in the text. “Hertwig’s rule, also known as the ‘long axis rule’, states that a cell is most likely to divide along its longest interphase axis, and has been observed in many cell types dividing within a two-dimensional plane (Bosveld et al., 2016; Thery and Bornens, 2006; Thery et al., 2005; Wyatt et al., 2015). In the epidermis, where basal cells divide in three-dimensional space, a cell’s longest axis may lie perpendicular to the epithelial plane, which could promote perpendicular spindle alignment and asymmetric division. Indeed, the perpendicular divisions of early *Xenopus* and zebrafish embryos have been shown to divide according to a three-dimensional long-axis rule (Chalmers et al., 2003; Xiong et al., 2014).”

Subsection “Cell division orientation is correlated with basal cell geometry”: "Basal cells in Vangl2^Lp/Lp^ embryos were packed more densely into the basal layer compared to wildtype (258 cells vs 215 cells per 1200μm2 field of view) (Figure 4E), but this increase in cell density was not due to elevated proliferation (Figure 4F).". Small difference in the proliferation rate can have a substantial impact on cell numbers at long time scale. Please reformulate.

We have softened this statement, which now reads “Basal cells in *Vangl2^Lp/Lp^* embryos were packed more densely into the basal layer compared to wildtype (258 cells vs 215 cells per 1200μm^2^ field of view) (Figure 4E). However, we did not detect elevated proliferation in *Vangl2^Lp/Lp^* embryos (Figure 4F).

Subsection “Cell division orientation is correlated with basal cell geometry”: "We also observed a modest but significant relationship between division plane and height:width aspect ratio along the apical-basal axis, wherein cells that divided in a planar orientation tended to have smaller height:width ratios (average = 0.8307) compared to cells that divided perpendicularly (average height:width ratios = 1.275) (Figure 4I). This relationship was also observed in Vangl2 mutant embryos (Figure 4H, J, Figure 4—video 2), where perpendicular divisions were associated with larger height:width ratios". The authors need to plot cell division orientation versus height: width ratio to substantiate this conclusion.

We have added Figure 4—Figure supplement 2 showing cell division orientation vs H:W ratio for wild type and *Vangl2^Lp/Lp^*embryos. The plots show a positive correlation between H:W ratio and division angle (0-90 degree) for both genotypes (correlation coefficient r=0.62 for wt and r=0.57 for *Vangl2^Lp/Lp^*).

Subsection “Spatial differences in epithelial packing and shape correlate with cell division orientations and the timing of stratification”: Are the orientation distributions statistically different in the lateral, intermediate and midline regions?

They are. We now include a statistical analysis of orientation distributions between different skin regions and find that in WT all three regions are significantly different but in the *Vangl2^Lp/Lp^*mutants, they are not. These data are now reported in Figure 7, Figure 7—figure supplement 1, and in their accompanying legends.

Subsection “Spatial differences in epithelial packing and shape correlate with cell division orientations and the timing of stratification”: "Thus, coverage of the epidermis over the spinal cord induces a gradient of cell density and elongation along the mediolateral axis, which in turn regulates cell division planes and the timing of stratification." Please weaken this conclusion.

The revised statement now reads, “Thus, coverage of the epidermis over the spinal cord induces a gradient of cell density and elongation along the mediolateral axis, which in turn regulates cell division planes across the epidermis.”

Subsection “Exogenous stretching of skin explants increases the frequency of planar cell divisions”: The 14.5 control and stretched conditions should be done in wt conditions. Accordingly, the conclusion at the end of the Results section is incorrect. The authors can only make this conclusion in a Vangl2 mutant conditions.

We apologize for not clearly articulating the rationale for the experimental design of the stretch experiments. Because wild type division planes are already strongly biased toward planar divisions at E14.5, we reasoned that stretching explants of this stage would be unlikely to have a quantifiable effect, and if anything would shift division planes to a more E13.5-like distribution, which would not have been an appropriate comparison for Vangl2 mutants. We specifically wanted to test whether stretch could rescue the planar division orientations of Vangl2 mutants such that their division planes would now resemble unstretched wild type skins. Thus, the appropriate control in this experiment is unstretched mutant skins. To determine the effect of stretch on wild type skins, we chose to stretch E15.5 when division planes are biased toward oblique and perpendicular orientations and asked whether stretch could shift that bias toward more planar orientations. Our results indicate that division planes are not fixed in either genotype, and that basal cells are able to shift division planes in response to changes in shape/tension.

We have substantially revised the text accompanying Figure 8 to better explain the experimental rationale, design, and conclusions of this experiment.